# Text-Aware Diffusion for Policy Learning

**Calvin Luo**   **Mandy He**[*]   **Zilai Zeng**[*]   **Chen Sun**
Brown University
{calvin_luo,mandy_he,zilai_zeng,chensun}@brown.edu

## Abstract

Training an agent to achieve particular goals or perform desired behaviors is often accomplished through reinforcement learning, especially in the absence of expert demonstrations. However, supporting novel goals or behaviors through reinforcement learning requires the ad-hoc design of appropriate reward functions, which quickly becomes intractable. To address this challenge, we propose Text-Aware Diffusion for Policy Learning (TADPoLe), which uses a pretrained, frozen text-conditioned diffusion model to compute dense zero-shot reward signals for text-aligned policy learning. We hypothesize that large-scale pretrained generative models encode rich priors that can supervise a policy to behave not only in a text-aligned manner, but also in alignment with a notion of naturalness summarized from internet-scale training data. In our experiments, we demonstrate that TADPoLe is able to learn policies for novel goal-achievement and continuous locomotion behaviors specified by natural language, in both Humanoid and Dog environments. The behaviors are learned zero-shot without ground-truth rewards or expert demonstrations, and are qualitatively more natural according to human evaluation. We further show that TADPoLe performs competitively when applied to robotic manipulation tasks in the Meta-World environment, without having access to any in-domain demonstrations.

## 1   Introduction

Can we train reinforcement learning agents that drive humanoids in a virtual environment [39] to stably stand? How about standing with *hands on hips*, *kneeling*, or *doing splits*? While state-of-the-art algorithms have shown success on the former scenario (e.g. [14]), the latter (illustrated in Figure 1) remains challenging due to the need for carefully (and often manually) crafted reward functions to specify the desired behaviors. The dependence on ad-hoc designed reward functions renders inscalable the learning of ever-increasing amounts of novel behaviors, which are required in applications ranging from character animation [2] to robotic manipulation [42].

Our work looks towards natural language as a powerful interface through which humans can flexibly specify desired goals or behaviors of interest. We therefore investigate how to construct a zero-shot text-conditioned reward signal, replacing the need for ad-hoc designs, through which text-aligned policies can be learned. We present Text-Aware Diffusion for Policy Learning (TADPoLe), which utilizes a large-scale pretrained, *frozen* text-conditioned diffusion model to generate a *dense* reward signal for policy learning. We hypothesize that generative diffusion models, which are pretrained on internet-scale datasets to produce text-aligned, natural-looking images [32, 29] and videos [3, 10, 15], can be utilized to automatically craft a *multimodal* reward signal that encourages an agent to behave both *faithfully* with respect to text conditioning and *naturally* with respect to human perception. Our method is novel in its reward computation, as well as its utilization of a domain-agnostic generative model, rather than one trained from environment-specific or task-specific video demonstrations, as used in prior work [9, 24, 6, 8, 20, 19].

---

[*]Equal contribution.

38th Conference on Neural Information Processing Systems (NeurIPS 2024).

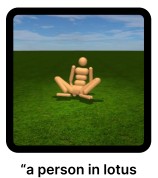 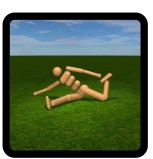 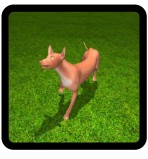 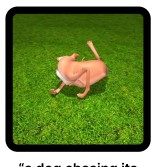 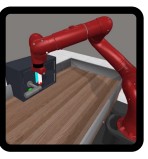 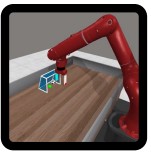

"a person in lotus position"  "a person doing splits"  "a dog standing"  "a dog chasing its tail"  "a robot arm is closing the door of a black safe"  "a robot arm is pushing a soccer ball into the net"

Figure 1: Our proposed Text-Aware Diffusion for Policy Learning (TADPoLe) framework leverages frozen, pretrained text-aware diffusion models to automatically craft dense text-conditioned rewards for policy learning. Here we visualize TADPoLe achieving diverse text-conditioned goals in the Humanoid, Dog, and Meta-World environments.

TADPoLe is motivated by the insight that a reinforcement learning policy can be viewed as an agent-centric *implicit* video representation when operating within an environment with visual rendering capabilities. As illustrated in Figure 2 (left), an agent's video generation process involves the selection of actions following a policy $\pi_\theta$, and the conversion of the action sequence into video subsequences through the environment's rendering function. A policy can therefore be seen as iteratively generating frames conditioned on the actions it selects; on the other hand, a text-to-image diffusion model can also be seen as generating static image frames, but conditioned on natural language instead. A connection can then be established between a policy and a diffusion model, where the frame or video segment "generated" by the policy can be critiqued by evaluating how likely a text-conditioned diffusion model would generate the same visuals, thus providing dense text-aligned reward signals to guide policy learning (Figure 2 right). Our work is inspired by DreamFusion [25], where a text-conditioned 3D model is learned through rendered views, and where volumetric raytracing ensures spatial consistency. Here, we seek to learn a text-conditioned policy through rendered frames or subsequences, where the environment naturally ensures temporal continuity and consistency with respect to a notion of physics instantiated by the environment.

Concretely, TADPoLe achieves text-conditioned policy learning by using a generative diffusion model in a discriminative manner. It computes the reward signal as a weighted combination of two reward terms, which aim to measure the alignment between the rendered observation and text conditioning, and the naturalness of the agent's behaviors, respectively. In this way, we can in effect "distill" the natural visual and motion priors as well as vision-text alignment understanding captured within the diffusion model into a policy. By default, TADPoLe uses a text-to-image diffusion model [31] to densely compute a reward signal solely from the immediate subsequent frame after each action. We then generalize the framework to Video-TADPoLe, which uses a text-to-video diffusion model [12] to calculate dense rewards as a function of a sliding context window of past as well as future frames achieved. The agent is thus trained to select actions such that arbitrary consecutive subsequences of frames are well-aligned with text as well as natural video (e.g. motion) priors.

We highlight TADPoLe as the first approach to leverage *domain-agnostic* visual generative models for policy learning. Through quantitative and human evaluations on Humanoid [39], Dog [39], and Meta-World [42] environments, we demonstrate that TADPoLe enables the learning of novel, zero-shot policies that are flexibly and accurately conditioned on natural language inputs, across multiple robot configurations and environments, for both goal-achievement and continuous locomotion tasks. TADPoLe therefore provides two main benefits simultaneously: a performant approach towards zero-shot policy learning, where complex reward functions no longer need to be manually specified per task, and a promising path towards distilling priors summarized from large-scale pretraining into policies, ultimately resulting in the learning of more naturally-aligned behavior within arbitrary environments. Visualizations and code are provided at diffusion-supervision.github.io/tadpole/.

## 2 Related Work

Diffusion models [34, 35, 16, 36] have recently demonstrated amazing generative modeling capabilities, particularly in the domain of text-conditioned image generation [5, 29, 32, 31]. Notably, guidance [36, 5, 17] has been shown to be a critical component in producing visual outputs aligned with textual data, enabling the generation of images that accurately match a desired text caption, especially when the models are scaled to utilize large foundation models such as CLIP [27] or T5 [28], and trained on massive image-text datasets [33]. Our work is inspired by DreamFusion [25], where a

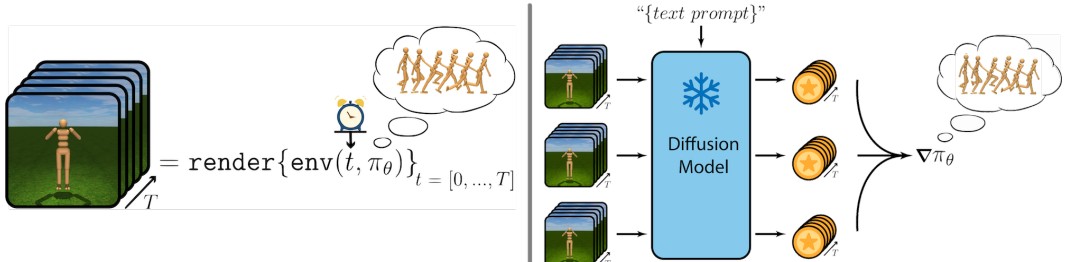

Figure 2: A policy $\pi_\theta$ that interacts with an environment can be treated as an agent-centric *implicit* video representation, where the arrow of time is actuated by the agent's actions and the pixels are rendered by the environment. The rendered behaviors can then be evaluated by a text-aware diffusion model to produce dense rewards, thereby providing text-conditioned update signals to the policy.

pretrained, frozen text-to-image diffusion model is able to supervise the learning of zero-shot 3D models conditioned on text. We propose leveraging pretrained diffusion models to supervise the learning of flexible, text-conditioned policies in a zero-shot manner over the time dimension.

There are numerous works that investigate how interactive agents can learn to perform behaviors specified by textual inputs. SayCan [1] grounds the knowledge of complex, high-level behaviors within an LLM to the context of a robot through pretrained behaviors. This then enables an LLM to instruct and guide a robot, through combining low-level behaviors, to perform complex temporally extended behaviors. LangLfP proposes a method for incorporating free-form natural language conditioning into imitation learning by first associating goal images with text captions and training a policy to follow either language or image goals, but only conditioning on natural language during test time inference [4]. The Text-Conditioned Decision Transformer learns a causal transformer to autoregressively produce actions conditioned on both text tokens as well as state and action tokens [26]. Similarly, Hiveformer proposes a unified multimodal Transformer model for robotic manipulation that conditions on natural language instructions, camera views, as well as past actions and observations [11]. However, LangLfP, Text-Conditioned Decision Transformer, and Hiveformer, all require training on datasets of trajectories that have been labelled with natural language. In contrast, TADPoLe enables the learning of text-conditioned policies irrespective of visual environment, and without requiring any pretraining dataset of demonstrations or labeling.

Similar to our work, UniPi [6] treats the sequential decision-making problem as a text-conditioned video generation problem. The authors propose training a video diffusion model to produce a future visual plan for the agent; the subsequent frames are then converted to actions by means of an inverse dynamics model. VLP [7] utilizes text-to-video generative models for planning and goal generation for an agent. Both methods require the video generative models to be trained *ad-hoc* on the target environments, whereas we directly use frozen general-purpose generative models. Mahmoudieh et al. [21] propose a framework that uses CLIP to generate a reward signal from a text description of a goal state and raw pixel observations from the environment, which is then used to learn a task policy. In VLM-RM [30], the authors also explore utilizing CLIP as the reward model for training humanoids to accomplish complex goal-reaching tasks. In our work, we investigate locomotion tasks on top of goal-achievement, and explore how using diffusion models to produce a reward signal can outperform CLIP-based approaches. Although not conditioned on text, VIPER [8] also aims to harness recent advancements in generative modeling by employing a video prediction model's likelihoods as a reward signal. However, VIPER does not enable the learning of policies conditioned on text, and requires in-domain expert videos for ad-hoc training the video model. Finally, Diffusion Reward [18] also extracts a reward from a diffusion model to train policies; however, it requires training an ad-hoc video model on expert trajectories, the collection of which cannot always be assumed to be trivial, and does not enable text-conditioned policy learning.

## 3 Method

We propose Text-Aware Diffusion for Policy Learning (**TADPoLe**) to learn text-aligned policies by leveraging frozen, pretrained text-conditioned diffusion models. An overview of the framework can be found in Figure 3.

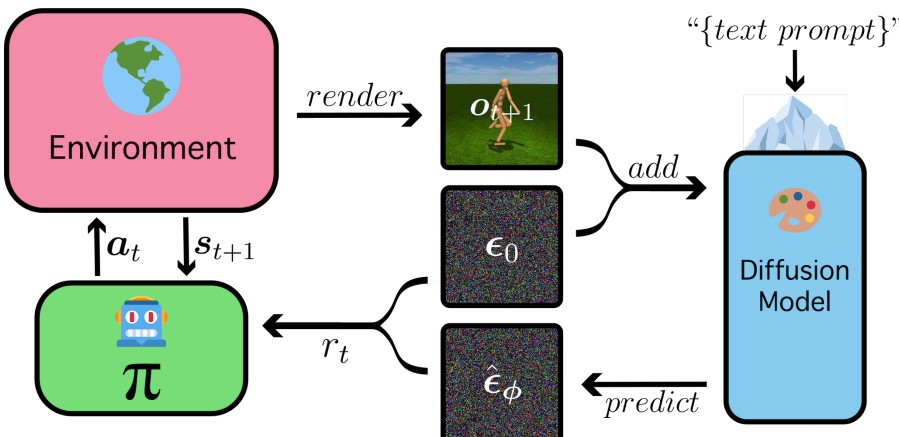

Figure 3: An illustration of the TADPoLe pipeline, which computes text-conditioned rewards for policy learning through a pretrained, frozen diffusion model. At each timestep, the subsequent frame rendered through the environment is corrupted with a sampled Gaussian source noise vector $\epsilon_0$. The pretrained text-conditioned diffusion model then predicts the source noise that was added. The reward is designed to be large when the selected action produces frames well-aligned with the text prompt.

## 3.1 Text-Aware Diffusion for Policy Learning

We first describe how TADPoLe produces text-conditioned rewards from image observations. At each timestep $t$, reward $r_t$ is computed as a score between rendered subsequent image $\mathbf{o}_{t+1}$ and the provided text caption describing the behavior of interest, denoted by $y$, using a frozen, pretrained text-to-image diffusion model. We begin by corrupting the rendered image $\mathbf{o}_{t+1}$ with a sampled Gaussian source noise vector $\epsilon_0 \sim \mathcal{N}(\epsilon; \mathbf{0}, \mathbf{I})$ to produce noisy observation $\tilde{\mathbf{o}}_{t+1}$, and use the diffusion model to make an unconditional prediction $\hat{\epsilon}_\phi(\tilde{\mathbf{o}}_{t+1}; \mathsf{t}_{\text{noise}})$ as well as a conditional prediction $\hat{\epsilon}_\phi(\tilde{\mathbf{o}}_{t+1}; \mathsf{t}_{\text{noise}}, y)$. Here $\hat{\epsilon}_\phi(\cdot)$ is a neural network that predicts the source noise given $\tilde{\mathbf{o}}_{t+1}$, the level of noise corruption $\mathsf{t}_{\text{noise}}$, and optionally the text prompt $y$; we overload notation to have $\hat{\epsilon}_\phi$ represent the source noise prediction in Figure 3. We then compute the mean squared error (MSE) between the two predictions as a reward signal $r_t^{\text{align}}$ to be *maximized*:

$$r_t^{\text{align}} = \|\hat{\boldsymbol{\epsilon}}_\phi(\tilde{\boldsymbol{o}}_{t+1}; \mathsf{t}_{\text{noise}}, y) - \hat{\boldsymbol{\epsilon}}_\phi(\tilde{\boldsymbol{o}}_{t+1}; \mathsf{t}_{\text{noise}})\|_2^2 .$$

As investigated in Appendix B.3, we empirically observe that $r_t^{\text{align}}$ plays a crucial role on the success of TADPoLe. We hypothesize that for an appropriately-selected noise corruption level $\mathsf{t}_{\text{noise}}$, this term measures the alignment between the environmental observation and the text prompt. Intuitively, for unconditional prediction $\hat{\boldsymbol{\epsilon}}_\phi(\tilde{\boldsymbol{o}}_{t+1}; \mathsf{t}_{\text{noise}})$, the model is incentivized only to bring the noisy input to any arbitrary cleaner image, and makes minimal edits by moving it towards the closest clean mode in data space. On the other hand, if the model recognizes visual features in the noisy image aligned with the text prompt, conditional prediction $\hat{\boldsymbol{\epsilon}}_\phi(\tilde{\boldsymbol{o}}_{t+1}; \mathsf{t}_{\text{noise}}, y)$ is incentivized to do "extra work" and bring it closer to the specific mode described by the text conditioning. We thus expect the MSE to be larger for well-aligned text conditioning. For an unaligned text prompt, the model may have more difficulty in recognizing relevant visual features in the corrupted image, and therefore generally has a lower computed $r_t^{\text{align}}$ signal. Therefore maximizing $r_t^{\text{align}}$ is a tractable proxy for maximizing the alignment between the rendered observation $\mathbf{o}_{t+1}$ and the provided text prompt $y$.

We also wish to encourage behaviors that are natural to human perception (e.g. a humanoid should walk similar to how a typical pedestrian would walk). We approximate the *naturalness* of a behavior by how accurately the diffusion model is able to predict the exact source noise vector that was applied. Intuitively, if it voluntarily predicts the exact noise vector with informative text conditioning, thereby perfectly reconstructing the query image, then the diffusion model believes the original rendered frame is reasonably natural (according to the priors captured by the diffusion model). We would therefore like to minimize $\|\hat{\boldsymbol{\epsilon}}_\phi(\tilde{\boldsymbol{o}}_{t+1}; \mathsf{t}_{\text{noise}}, y) - \boldsymbol{\epsilon}_0\|_2^2$. We would also like this term to be comparatively closer to the source noise vector than the unconditional prediction is, further reaffirming the benefit of the text conditioning. We therefore seek to also *maximize* a comparative reconstruction term as below:

$$r_t^{\text{rec}} = \|\hat{\boldsymbol{\epsilon}}_\phi(\tilde{\boldsymbol{o}}_{t+1}; \mathsf{t}_{\text{noise}}) - \boldsymbol{\epsilon}_0\|_2^2 - \|\hat{\boldsymbol{\epsilon}}_\phi(\tilde{\boldsymbol{o}}_{t+1}; \mathsf{t}_{\text{noise}}, y) - \boldsymbol{\epsilon}_0\|_2^2 .$$

**Algorithm 1** Text-Aware Diffusion for Policy Learning (TADPoLe)

```
 1: prompt = sample(action_phrase)
 2: π_θ = initialize(θ)
 3: D ← {}
 4: while not converged:
 5:        s_0 ∼ p(s_0)
 6:        for t in range(episode_length):
 7:              a_t ∼ π_θ(a_t | s_t)
 8:              s_{t+1} ∼ P(s_{t+1} | s_t, a_t)
 9:              ε_0 ∼ N(ε; 0, I)
10:              o_{t+1} ∼ P(o_{t+1} | s_{t+1})
11:              õ_{t+1} ← noisify(o_{t+1}, ε_0, t_noise)
12:              r_t = tadpole_reward(õ_{t+1}, ε_0, prompt)
13:              τ ← τ ∪ (s_t, a_t, r_t, s_{t+1})
14:        D ← D ∪ τ
15:        loss = policy_loss(D)
16:        grads = gradient(loss, θ)
17:        opt.apply_gradients(grads, θ)
```

Ultimately, we compose these two terms into a final reward signal $r_t$ exposed to the policy during training. We scale each of the individual terms with tunable hyperparameter constants, and apply a `symlog` [13] transformation operation:

$$r_t = \texttt{symlog}(w_1 * r_t^{\text{align}}) + \texttt{symlog}(w_2 * r_t^{\text{rec}}).$$

The choice of using `symlog` as a reward normalization technique is thoroughly studied in Section 4.5. TADPoLe is agnostic to the specific choices of policy network architecture and optimization objectives. A pseudocode of the method is provided in Algorithm 1. It is worth emphasizing that $t_{\text{noise}}$ and subscript-less $t$ refer to different notions of time; $t$ indexes the timestep of the agent in the environment, whereas $t_{\text{noise}}$ determines the level of noise to corrupt the raw observed image.

## 3.2 TADPoLe with Text-to-Video Diffusion Models

Conceptually, there exist fundamental limitations to using a text-to-image model to provide a reward signal. As each image is evaluated statically and independently, we are unable to expect the text-to-image diffusion model to be able to accurately understand and supervise an agent in learning notions of speed, or in some cases, direction, as such concepts require evaluating multiple consecutive timesteps to deduce. We therefore propose Video-TADPoLe, where a dense text-conditioned reward signal is calculated over sliding windows of consecutive frames through a pretrained text-to-video diffusion model. We extend and generalize the reward formulation from TADPoLe thusly.

We can compute reward terms for arbitrary start index $i$ and end index $j$ inclusive, for $i \leq j$, by considering the sequence of subsequently rendered frames $\mathbf{o}_{[i+1:j+1]}$. We once again utilize source noise vector $\boldsymbol{\epsilon}_0 \sim \mathcal{N}(\boldsymbol{\epsilon}; \mathbf{0}, \mathbf{I}_{j-i+1})$ to produce noisy observation $\tilde{\mathbf{o}}_{[i+1:j+1]}$. Then, we can compute a batch of alignment reward terms through one inference step of the text-to-video diffusion model as:

$$r_{[i:j]}^{\text{align}} = \left\| \hat{\boldsymbol{\epsilon}}_\phi(\tilde{\boldsymbol{o}}_{[i+1:j+1]}; t_{\text{noise}}, y) - \hat{\boldsymbol{\epsilon}}_\phi(\tilde{\boldsymbol{o}}_{[i+1:j+1]}; t_{\text{noise}}) \right\|_2^2,$$

and a batch of reconstruction reward terms as:

$$r_{[i:j]}^{\text{rec}} = \left\| \hat{\boldsymbol{\epsilon}}_\phi(\tilde{\boldsymbol{o}}_{[i+1:j+1]}; t_{\text{noise}}) - \boldsymbol{\epsilon}_0 \right\|_2^2 - \left\| \hat{\boldsymbol{\epsilon}}_\phi(\tilde{\boldsymbol{o}}_{[i+1:j+1]}; t_{\text{noise}}, y) - \boldsymbol{\epsilon}_0 \right\|_2^2.$$

For a desired context window of size $n$, we then calculate the reward at each timestep $t$ utilizing each context window that involves achieved observation $\mathbf{o}_{t+1}$:

$$r_t = \frac{1}{n} \sum_{i=1}^{n} \texttt{symlog}\left( w_1 * r_{[t-i+1:t-i+n]}^{\text{align}}[i-1] \right) + \texttt{symlog}\left( w_2 * r_{[t-i+1:t-i+n]}^{\text{rec}}[i-1] \right).$$

Intuitively, we seek to calculate an overall reward for an action based off how well the resulting rendered frame aligns with text-conditioning at the beginning of a motion sequence, the end of one, and arbitrarily inbetween. For window size $n = 1$, this recreates TADPoLe behavior, but using a text-to-video model; for $n > 1$, we make the computation tractable through dynamic programming.

# 4 Experiments

We now demonstrate the effectiveness of TADPoLe on goal achievement, continuous locomotion, and robotic manipulation tasks. All results are achieved without access to in-domain demonstrations.

## 4.1 Experimental Setup and Evaluation

**Benchmarks:** We present our main results using the Dog and Humanoid environments from the DeepMind Control Suite [39], and robotic manipulation tasks from Meta-World [42]. Dog and Humanoid are known to be challenging due to their large action space, complex transition dynamics, and lack of task-specific priors (such as termination conditions). We update the environments by modifying the terrain rendered by MuJoCo [38] to have green grass and blue sky. We also limit the number of environment timesteps to be 300, which is sufficient to demonstrate successful learning of a behavior, rather than the default 1000. The agent's initialized joint configurations are also fixed, as we focus on learning text-conditioned capabilities rather than robustness to initialization conditions. Meta-World was initially designed for multi-task and meta-reinforcement learning, and was later adopted to evaluate language-conditioned imitation learning algorithms [23, 22, 37]. We select a suite of tasks which are balanced for diversity and complexity, and pair each task with a text prompt (see tasks and their corresponding prompts in Appendix C). Following prior design [18] for Meta-World, we also add a sparse success signal to the dense text-conditioned reward signals.

**Implementation:** We use TD-MPC [14] as the reinforcement learning algorithm for all tasks. It is the first documented model to solve Dog tasks when ground truth rewards are available for walking. We fix the hyperparameters to the default ones recommended by the TD-MPC authors (see Table A3 in Appendix) for all experiments unless otherwise mentioned. We train Humanoid and Dog agents for 2M steps, and Meta-World agents for 700K steps. For Meta-World experiments, we scale the sparse success signal by 2. Visualizations and quantitative evaluations are reported using the *last* checkpoint achieved at the end of training. We use StableDiffusion 2.1 [31] as the text-to-image diffusion model ($\sim$1.3B parameters), and AnimatedDiff [12] v2 ($\sim$1.5B parameters) as the text-to-video diffusion model. AnimatedDiff is implemented on top of StableDiffusion 1.5. We fix the reward weights $w_1 = 2000$ and $w_2 = 200$ based on Humanoid standing and walking performance, and study their impact in Appendix B.3. Selection of noise level is discussed in Appendix A. All experiments are performed on a single NVIDIA V100 GPU.

**Baselines:** We compare TADPoLe against other text-to-reward approaches, including VLM-RM [30], LIV [20], and Text2Reward [41], on top of the same underlying TD-MPC architecture, hyperparameters, and optimization scheme for fair comparison. For LIV, we use their provided CLIP-based checkpoint finetuned on robotic demonstration videos. For VLM-RM, we utilize the ViT-bigG-14 CLIP checkpoint ($\sim$1.3B parameters), reported as the best performing in their work. We follow a prompt template provided in the Text2Reward paper to generate reward functions for the Dog and Humanoid agent, interfaced through vanilla ChatGPT using GPT-3.5. Whereas VLM-RM and LIV provide a multimodal reward signal, and are more directly comparable to TADPoLe, it is of note that Text2Reward generates a text-conditioned reward function purely as the output of a pretrained language model. However, Text2Reward does have access to underlying sensor data such as speed and direction in real-time, whereas the visual interface approaches, including TADPoLe, do not.

**Evaluation Protocols:** We benchmark all text-conditioned methods with a corresponding standardized prompt for fair comparison, and report both quantitative as well as qualitative comparisons. We use cumulative ground-truth rewards as quantitative evaluation metrics for Dog and Humanoid when it is available. We note this is a naturally *unfavorable* comparison for methods that provide a text-conditioned reward signal purely through a visual interface, as the reward the agent receives has no access to the underlying sensors (such as ones that measure speed and energy usage) that the ground-truth reward function uses to evaluate performance. For example, the ground-truth reward function may have an arbitrary threshold on a speed sensor that needs to be hit to constitute successful "walking", and a separate threshold for "running"; however the detailed characteristics of and even existence of such a sensor, as well as any thresholds surrounding it, are hidden for policies supervised only through vision and language feedback. Nonetheless, it offers a standardized, numerical comparison across all methods. For Meta-World, we report the "success rate" evaluation metric, computed as the proportion of evaluation rollouts in which the agent successfully completes the given task.

Table 1: Results for goal-achievement experiments on DeepMind Control Suite Dog and Humanoid environments. For rows with an associated ground-truth reward function, numerical results are listed; for performant approaches, we report mean and standard deviation across 5 seeds. For novel zero-shot text-conditioned behavior learning, checkmarks denote if the resulting policy is aligned with the provided text prompt according to human evaluation.

| Environment | Prompt | VLM-RM | LIV | Text2Reward | TADPoLe (Ours) | Ground-Truth |
|---|---|---|---|---|---|---|
| Humanoid | "a person standing" | 247.05 (± 16.90) | 11.27 | 10.50 | 254.43 (± 8.76) | 287.68 (± 4.64) |
| Humanoid | "a person in lotus position" | ✔ | ✘ | ✘ | ✔ | – |
| Humanoid | "a person doing splits" | ✔ | ✔ | ✘ | ✔ | – |
| Humanoid | "a person kneeling" | ✘ | ✘ | ✘ | ✔ | – |
| Dog | "a dog standing" | ✘ | ✘ | ✘ | ✔ | – |

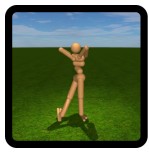
"a person standing"

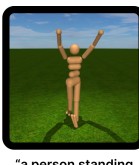
"a person standing with hands above head"

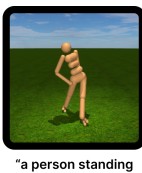
"a person standing with hands on hips"

Figure 4: TADPoLe demonstrates sensitivity to subtle variations to the input prompt, learning to stand in different positions with only slight modifications to the text conditioning.

| Prompt | TADPoLe Naturalness (↑) |
|---|---|
| "a dog standing" | 87.5% |
| "a person in lotus position" | 62.5% |
| "a person doing splits" | 62.5% |
| "a person kneeling" | 70.8% |
| "a person walking" | 84.0% |
| "a dog walking" | 76.0% |

Table 2: Qualitative study: percentages denote user preference for the naturalness of the resulting motion produced by TADPoLe over VLM-RM (goal-achievement) and Video-TADPoLe over ViCLIP-RM (continuous locomotion).

A main benefit of utilizing a reward signal conditioned flexibly on text is the ability to learn policies with behaviors beyond those defined by existing ground-truth reward functions. As these have no corresponding ground-truth reward functions, quantitative comparison across different text-conditioned methods is challenging; we therefore appeal to a qualitative user study. We perform a paid study through the Prolific platform, with a total of 25 anonymous random participants without prior training to estimate a general response from the human population. For a video demonstration from each trained model, selected as the last timestep of policy training without cherry-picking, each participant is asked if it sufficiently aligns with the text prompt it was conditioned on. These results are depicted in tables as checkmarks (✔) and x-marks (✘), where a checkmark denotes if a majority of participants believe it is text-aligned. In Table A7, we provide the fine-grained user study results on what percentage of the users believe the video achieved by the policy is appropriately text-aligned. We then proceed with a user study regarding naturalness. Given a video produced by VLM-RM and TADPoLe, users are given a choice as to which they believed to be the more natural motion or pose. This seeks to approximate how naturally the resulting Humanoid and Dog policies behave, according to human belief over how people and dogs naturally move in the real world.

## 4.2 Goal Achievement

For text-conditioned goal-achievement, the objective is to learn a policy to consistently achieve a particular pose described by a text prompt; as the emphasis is for every frame to match a fixed goal pose rather than performing continuous motion, it is natural to apply TADPoLe with text-to-image diffusion models. We set the noise level $t_{noise} \sim U(400, 500)$, with intuition provided in Appendix A.

In the Humanoid environment, there is a ground-truth reward function that measures standing performance, as a function of the straightness of the agent's spine. We therefore compare all text-conditioned methods using the provided reward function as a quantitative metric, with a standard prompt of "a person standing"; these results are shown in the first row of Table 1. TADPoLe and VLM-RM achieve competitive quantitative performance with an agent trained on the ground-truth reward function. The following rows show that according to the user study, TADPoLe consistently achieves text-aligned behaviors beyond making the Humanoid stand, whereas other approaches often fail. Table 2 shows that users consistently found TADPoLe to produce more natural-looking motions and poses when compared head-to-head with VLM-RM.

Table 3: Results for continuous locomotion experiments. For rows with an associated ground-truth reward function, numerical results are listed; for performant approaches, we report mean and standard deviation across 5 seeds. Evaluation for text-alignment is also reported. Video-TADPoLe greatly outperforms ViCLIP-RM on both Humanoid and Dog.

| Environment | Prompt | LIV | Text2Reward | Video-TADPoLe (Ours) | ViCLIP-RM | Ground-Truth |
|---|---|---|---|---|---|---|
| Humanoid | "a person walking" | 0.65 | 3.35 | 145.60 ($\pm$ 48.20) | 25.51 ($\pm$ 40.45) | 275.06 ($\pm$ 9.21) |
| Dog | "a dog walking" | 16.86 | 63.15 | 60.20 ($\pm$ 8.82) | 14.67 ($\pm$ 9.84) | 280.07 ($\pm$ 3.07) |
| Humanoid | "a person walking" | ✘ | ✘ | ✔ | ✔ | ✔ |
| Dog | "a dog walking" | ✘ | ✘ | ✔ | ✘ | ✔ |

We then investigate whether or not TADPoLe is sensitive to subtle variations of the input prompt. We change the text prompt from "a person standing" to "a person standing with hands above head" and "a person standing with hands on hips". In Figure 4, we visually verify that the resulting Humanoid policy can indeed learn distinct behaviors that respect the different hand placement specifications. We take this as evidence that TADPoLe is capable of respecting fine-grained details and subtleties of the input prompts when learning text-conditioned policies.

## 4.3 Continuous Locomotion

We further explore the ability of TADPoLe to learn continuous locomotion behaviors conditioned on natural language specifications. Such tasks are often difficult to learn purely from static external description, as there is no canonical pose or goal frame that if reached, would denote successful achievement of the task. This is challenging for approaches that statically select a canonical goal-frame to achieve, such as CLIP or LIV, and we propose Video-TADPoLe, which leverages large-scale pretrained text-to-video generative models, as a promising direction forward.

We utilize a noise level $t_{noise} \sim U(500, 600)$ in our continuous locomotion experiments. We perform a search over context windows of size $n = \{1, 2, 4, 8\}$, and report the best configuration per task. We observe that when the context window is too high (e.g. 8 or higher), the agent has consistently lower performance, and that although the agent learns coherent motion and repeats it, the pose is less text-aligned. For fair comparison against a text-video alignment model trained in a contrastive manner, we extend VLM-RM to ViCLIP-RM, where a ViCLIP-L-14 checkpoint [40] finetuned from ViT-L-14 CLIP is used to compute dense, text-conditioned rewards. At each timestep $t$, we compute dense rewards as cosine similarity between the encoded representations of video observation up to $t + 1$ and the text prompt. We ask ViCLIP to encode 8 video frames at a time, which is adopted by its authors for zero-shot experiments.

For the Humanoid task, we find that Video-TADPoLe achieves the best results amongst methods trained purely from visual and/or language feedback as in Table 3. On the other hand, ViCLIP-RM indeed learns to take steps, but does so sideways while maintaining an unnaturally lopsided pose. Meanwhile, LIV and Text2Reward fail to learn meaningful behaviors.

For the Dog task, we notice that the policy learned via ViCLIP-RM collapses; it learns to strike a particular pose and maintain it for perpetuity. Text2Reward, which does not have access to any visual information, but does have access to ground-truth state information for the Dog including speed, direction, and joint positions, achieves a reward of 63.15. Ultimately, Video-TADPoLe achieves a comparable result using a context window of 4, while also distinguishing itself as the most natural-looking policy in qualitative terms as the Dog agent appears to perform step-taking motions, rather than remain stationary. Table 2 further showcases a higher preference for the naturalness of the learned policies for continuous locomotion achieved by Video-TADPoLe compared to ViCLIP-RM, in both Dog and Humanoid environments.

In Figure 5, we visualize episode return curves achieved by Video-TADPoLe for a Humanoid agent with the prompt "a person walking". We visualize how during training, the computed Video-TADPoLe reward shares a positive correlation with the reward computed by a ground-truth function for walking over all episodes, lending confidence to it as a coherent, well-defined reward function. We also visualize the ground-truth evaluation curve. Figure A4 offers another example for "a dog walking".

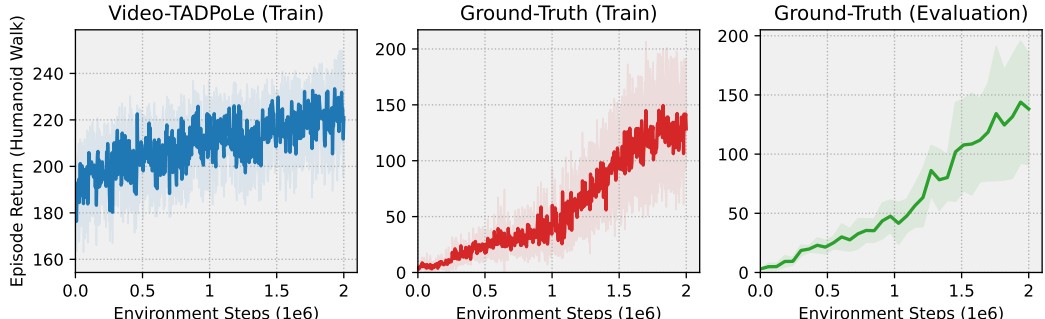

Figure 5: Episode return curves for a Humanoid agent trained with Video-TADPoLe, using the prompt "a person walking". We observe that the Video-TADPoLe reward signal (left) is positively correlated with the agent's performance as measured with ground-truth reward during training (middle) and evaluation (right). Shaded regions denote the standard deviation across five random seeds.

Table 4: Average success rate for robotic manipulation tasks in Meta-World [42] over 30 evaluation rollouts. We compare between TADPoLe and VLM-RM, both approaches that do not utilize in-domain data or demonstrations, and find TADPoLe significantly outperforms VLM-RM. We report mean performance and standard deviation across 10 seeds for each task.

| Success Rate (%) | Door Open | Door Close | Drawer Open | Drawer Close | Window Open | Window Close |
|---|---|---|---|---|---|---|
| VLM-RM | 0 ($\pm$ 0) | 79.7 ($\pm$ 39.8) | 10.0 ($\pm$ 30.0) | 100.0 ($\pm$ 0) | 9.7 ($\pm$29.0) | 0 ($\pm$ 0) |
| TADPoLe | 40.0 ($\pm$ 49.0) | 100.0 ($\pm$ 0) | 45.3 ($\pm$ 46.6) | 100.0 ($\pm$ 0) | 74.0 ($\pm$ 37.9) | 30.0 ($\pm$ 45.8) |

| Success Rate (%) | Coffee Push | Button Press | Soccer | Lever Pull | *Average* | |
|---|---|---|---|---|---|---|
| VLM-RM | 4.0 ($\pm$ 12.0) | 30.0 ($\pm$ 45.8) | 5.3 ($\pm$ 11.6) | 11.3 ($\pm$ 18.3) | 25.0 | |
| TADPoLe | 18.6 ($\pm$ 20.5) | 73.0 ($\pm$ 38.5) | 25.0 ($\pm$ 15.4) | 0 ($\pm$ 0) | **50.6** | |

## 4.4 Robotic Manipulation

We further investigate how well TADPoLe can be applied to learn robotic manipulation tasks through dense text-conditioned feedback. We do so by replacing the manually-designed ground-truth dense reward for each Meta-World task with TADPoLe's text-conditioned reward. Since TADPoLe aims to leverage domain-agnostic diffusion models for policy learning, we focus our evaluation to compare with baseline methods that also do not utilize in-domain (expert) demonstrations for the robotic manipulation tasks. We note that most of the prior methods which report performance on Meta-World rely on (often expert-produced) video demonstrations from a similar domain or the target environment directly for representation learning [23], reward learning [18], or both [20]. They are thus not directly comparable to TADPoLe.

We perform thorough comparisons between TADPoLe and VLM-RM by evaluating them on a diverse set of selected Meta-World tasks. Both models reuse the setup in Section 4.2 without modification, with training performed for 700k steps. In Table 4, we report the final success rate for each manipulation task averaged over 30 evaluation rollouts. We highlight that TADPoLe achieves high success rates across a variety of tasks, and significantly exceeds VLM-RM in terms of average overall performance. We take this as a positive signal that TADPoLe can meaningfully provide dense text-conditioned rewards that replace dense ground-truth hand-designed feedback. We also highlight how TADPoLe is able to successfully supervise the learning of policies within the synthetic-looking visual environment of Meta-World without finetuning the pretrained text-to-image diffusion model, despite the visual attributes (such as the appearance of the robotic arm, or the quality of the renderings) being quite dissimilar from the style of images StableDiffusion was trained on.

## 4.5 Normalization Study

Of interest is what reward normalization technique is most performant for adjusting the raw computed alignment and reconstruction terms into a final reward used for policy learning. We investigate a

Table 5: Quantitative results for TADPoLe and Video-TADPoLe with and without the `symlog` normalization operation, averaged over 3 seeds. Hyperparameters such as weights and noise level were kept the same across all experiments. We discover that not only does using the `symlog` transformation enable the highest empirical rewards, it also facilitates the reuse of hyperparameters across tasks, environments, and other diffusion models.

| Environment | Method | Prompt | SymLog | Direct Scaling | SymExp | Min-Max | Standardization |
|---|---|---|---|---|---|---|---|
| Humanoid | TADPoLe | "a person standing" | **267.23** | 239.74 | 241.49 | 256.81 | 236.59 |
| Humanoid | Video-TADPoLe | "a person walking" | **226.29** | 4.58 | 3.68 | 61.31 | 134.66 |
| Dog | Video-TADPoLe | "a dog walking" | **81.22** | 35.30 | 9.46 | 6.15 | 5.05 |

variety of normalization strategies in a quantitative manner in Table 5, reusing parameters $w_1 = 2000$ and $w_2 = 200$ across experiments, the selection of which is detailed in Section B.3 and Table A4.

Apart from the `symlog` transformation, we also compare against using no additional normalization (denoted as "Direct Scaling"), and using `symexp`. We also compare against empirical normalization techniques. This includes min-max rescaling, where an empirically estimated minimum value is subtracted from the achieved reward, which is then divided by an empirically calculated min-max range, and rescaled to $[-1, 1]$. This also includes standardization, which subtracts an empirically estimated mean from the achieved reward and then divides it by an empirically estimated standard deviation. We apply these techniques across Humanoid and Dog environments, for both TADPoLe and Video-TADPoLe. We discover that the `symlog` operation is the reward normalization strategy that achieves the best empirical results across robotic configurations, visual environments, and desired tasks, while reusing the same hyperparameter settings. We hypothesize that it helps to normalize the raw computed reward signals across diffusion models and environments to be roughly on the same scale. Indeed, we showcase how removing the `symlog` transformation, as well as using other normalization techniques, reduces consistent policy learning performance across environments and diffusion models with the same fixed hyperparameters. We further note that it does not require empirically estimated values, unlike min-max and standardization.

## 5 Conclusion and Future Work

We present Text-Aware Diffusion for Policy Learning (TADPoLe), a framework that optimizes a policy according to a provided natural language prompt through a pretrained text-conditioned diffusion model. TADPoLe enables novel behaviors to be learned in a zero-shot manner purely from text conditioning, and also offers a promising angle to train policies to behave in accordance with natural priors summarized from large-scale pretraining. TADPoLe can be applied across visual environments and different robotic states without modification, and we experimentally demonstrate that TADPoLe is able to learn novel goal-achievement as well as continuous locomotion behaviors conditioned only on text, across Humanoid, Dog, and Meta-World environments.

**Limitations:** An observed limitation of TADPoLe is that it is difficult to explicitly control the weight each individual word of an input text prompt has on the reward provided to the agent. For certain prompts, TADPoLe could potentially cause the agent to remain stationary since it may focus on alignment with the noun in the phrase rather than details of the goal. How to provide fine-grained control over the text-conditioning is an interesting direction to explore in future work. Further interesting future work includes utilizing multiple camera views simultaneously to compute the dense reward, as environments generally allow flexible rendering from arbitrary angles. Another observed limitation is that TADPoLe depends on a highly stochastic operation, namely, repeatedly resampling a Gaussian source noise vector at each timestep. The behavior of the resulting policy, after training for many iterations, can therefore vary for the same input text prompt, and potentially cause high variance in both visual and quantitative performance. How to control the stability of convergence to a consistent policy across repeated runs is an interesting future direction for exploration.

**Acknowledgements.** We would like to thank Amil Merchant, Daniel Ritchie, David Bau, Ben Poole, Ting Chen, Nate Gillman, and Yilun Du for helpful discussions and feedback. This work is supported by Samsung Advanced Institute of Technology, NASA, and a Richard B. Salomon Faculty Research Award for Chen Sun. Our research was conducted using computational resources at the Center for Computation and Visualization at Brown University.

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

# A Intuition Regarding A Reasonable Noise Level Range

We desire a reward signal that is high when the text prompt is well-aligned with the frame rendered following a well-selected action. However, we find that using too high or too low a noise level will cause the reward computation to ignore or discount the provided text prompt.

For a sufficiently low noise value, the conditional and unconditional noise prediction is similar for a particular text prompt and rendered frame that already has high cross-modal alignment. Being already close in data space to the desired mode described by the text prompt, the unconditional prediction will also cheaply seek to denoise the input towards the original input. Therefore, too low a noise value may cause the computed reward signal to decrease as cross-modal alignment increases, which runs counter to what is desirable. This is supported by looking at the left tails of the two graphs depicted in Figure A1, where the difference in computed TADPoLe rewards between well-aligned paired inputs and misaligned paired inputs is small.

Similarly, choosing a sufficiently high noise value may also cause unconditional and conditional predictions to be similar. For a given noisy input with virtually all of the spatial structure perturbed, the pretrained denoising model intuitively makes denoising predictions that fill in general structure to the image irrespective of text-conditioning. This is intuitively similar to the behavior of an unconditional prediction. Once again, we can visually observe this in the right tails of both graphs in Figure A1; for high noise values, both misaligned and well-aligned paired inputs have similar computed TADPoLe rewards.

Intuitively, TADPoLe is able to meaningfully quantify text-conditioned alignment with rendered observations into a reward signal for a noise level range approximately in the middle. For a balanced level of noise corruption on a text-aligned rendered observation, there is enough existing visual structure to help the denoising model make a meaningful text-conditioned prediction close to the original input, whereas an unconditional prediction may not do so accurately. In Figure A1, we observe that for a noise range in the middle, TADPoLe is able to substantially favor the well-aligned pair over the misaligned pair, respecting changes in both text prompt as well as rendered observations. We verify in our experiments that using a noise level between 400 and 500 works well for TADPoLe, across both Dog and Humanoid environments.

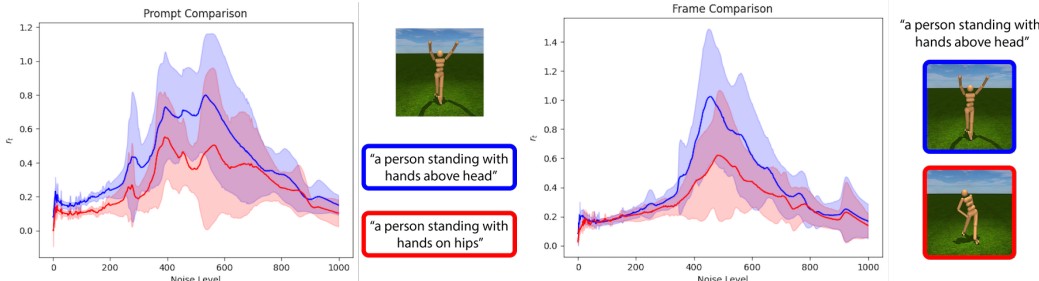

Figure A1: Noise range intuition for a fixed image but two distinct prompts (left), and for a fixed prompt but two distinct images (right). Through visualization, we verify that $U(400, 500)$ is a reasonable range from which to sample noise levels that can meaningfully distinguish vision-text alignment for arbitrarily rendered frames.

## A.1 Noise Level for Video-TADPoLe

We discover that Video-TADPoLe achieves better performance with a higher noise level than TADPoLe, and use a range of 500 to 600 in our experiments. We hypothesize that being able to observe multiple frames at once enables the space-time U-Net to exploit uncorrupted (or lesser-corrupted) portions from adjacent frames across the context window to inform how each individual frame should be denoised coherently. Because each frame in a context window has a distinct source noise applied to it, structural information can be leaked from randomly uncorrupted portions of the other frames in the context window when making predictions. Therefore, to make the denoising prediction more challenging, thereby forcing it to rely on the text-conditioning and learned motion priors rather than what is readily extractable from the provided context, generally a higher noise level is acceptable compared to TADPoLe.

# B    Detailed Hyperparameters

In our experiments, we place less emphasis on the underlying architecture and optimization scheme, instead comparing our method against other text-conditioned reward functions. We keep the same reinforcement learning architecture, optimization scheme, and text-to-image/text-to-video generative model in all environments, unless otherwise noted. We provide detailed hyperparameters about these existing components below.

## B.1    Pretrained Text-Conditioned Diffusion Models

We utilize a StableDiffusion v2.1 checkpoint for our TADPoLe experiments. We provide the sizes of the components in Table A1. For Video-TADPoLe experiments, we use a pretrained AnimateDiff checkpoint, and provide relevant details in Table A2. We do not update or modify either checkpoint during our training, utilizing them purely for inference.

## B.2    TD-MPC

We include the default hyperparameters from the TD-MPC implementation in Table A3 for completeness. We do not modify the default recommended settings for both Humanoid and Dog environments, as well as the Meta-World experiments.

Table A1: **StableDiffusion Components.** For completeness, we list sizes of the components of the StableDiffusion v2.1 checkpoint used in TADPoLe experiments. The checkpoint is used purely for inference, and is not modified or updated in any way. Note that the VAE Decoder is not utilized in our framework.

| Component | # Parameters (Millions) |
|---|---|
| VAE (Encoder) | 34.16 |
| VAE (Decoder) | 49.49 |
| U-Net | 865.91 |
| Text Encoder | 340.39 |

Table A2: **AnimateDiff Components.** For completeness, we list sizes of the components of the AnimateDiff checkpoint used in Video-TADPoLe experiments. The checkpoint is used purely for inference, and is not modified or updated in any way. Note that the VAE Decoder is not utilized in our framework.

| Component | # Parameters (Millions) |
|---|---|
| VAE (Encoder) | 34.16 |
| VAE (Decoder) | 49.49 |
| U-Net | 1312.73 |
| Text Encoder | 123.06 |

Table A3: **TD-MPC hyperparameters.** We use the official implementation TD-MPC [14] with no adjustments to the hyperparameters, but list it below for completeness. However, we do set the number of training steps to 2 million for all experiments using TD-MPC.

| Hyperparameter | Value |
|---|---|
| Discount factor ($\gamma$) | 0.99 |
| Seed steps | $5,000$ |
| Replay buffer size | Unlimited |
| Sampling technique | PER ($\alpha = 0.6, \beta = 0.4$) |
| Planning horizon ($H$) | 5 |
| Initial parameters ($\mu^0, \sigma^0$) | $(0, 2)$ |
| Population size | 512 |
| Elite fraction | 64 |
| Iterations | 12 (Humanoid) |
|  | 8 (Dog) |
| Policy fraction | 5% |
| Number of particles | 1 |
| Momentum coefficient | 0.1 |
| Temperature ($\tau$) | 0.5 |
| MLP hidden size | 512 |
| MLP activation | ELU |
| Latent dimension | 100 (Humanoid, Dog) |
| Learning rate | 3e-4 (Dog) |
|  | 1e-3 (Humanoid) |
| Optimizer ($\theta$) | Adam ($\beta_1 = 0.9, \beta_2 = 0.999$) |
| Temporal coefficient ($\lambda$) | 0.5 |
| Reward loss coefficient ($c_1$) | 0.5 |
| Value loss coefficient ($c_2$) | 0.1 |
| Consistency loss coefficient ($c_3$) | 2 |
| Exploration schedule ($\epsilon$) | $0.5 \to 0.05$ (25k steps) |
| Planning horizon schedule | $1 \to 5$ (25k steps) |
| Batch size | 2048 (Dog) |
|  | 512 (Humanoid) |
| Momentum coefficient ($\zeta$) | 0.99 |
| Steps per gradient update | 1 |
| $\theta^-$ update frequency | 2 |

## B.3    Selecting $w_1$ and $w_2$

We select hyperparameters $w_1$ and $w_2$, which scale the alignment term $r^{\text{align}}$ and the reconstruction term $r^{\text{rec}}$ respectively, firstly such that the resulting computed reward $r$ has value roughly between 0 and 1 at any arbitrary timestep. This is visualized in Figure A1, where over all noise levels, the

computed reward stays roughly between 0 and 1. Because the same pretrained text-conditioned diffusion model is used across all environments without modification, these hyperparameters can be generally reused to achieve the same kind of reward scale across environments. We found that the values of $w_1 = 2000$ and $w_2 = 200$ through a light hyperparameter sweep, reported in Table A4, and indeed verify that they work well without modification across both Humanoid and Dog environments in our main experiments.

Table A4: Ablation over $w_1$ and $w_2$ on Humanoid Stand and Walk

| $w_1$ | $w_2$ | Humanoid Stand | Humanoid Walk |
|-------|-------|----------------|---------------|
| 200   | 2000  | 249.82         | 2.72          |
| 1000  | 2000  | 240.05         | 121.20        |
| 2000  | 200   | **262.22**     | **226.29**    |
| 2000  | 1000  | 195.87         | 152.32        |
| 2000  | 2000  | 218.22         | 90.20         |

## C Meta-World Tasks

We visualize the complete suite of selected Meta-World tasks in Figure A2 along with the official names of the tasks. In Table A5 we also list the corresponding prompts consistently utilized across all text-conditioned approaches. In Table A6, we list the average success for all 12 robotic manipulation tasks. As we add a sparse "success" signal to text-conditioned approaches such as VLM-RM and TADPoLe, following prior experimental design [18] for Meta-World, we also list the performance of utilizing the sparse signal only.

We first notice that no approach is able to solve two selected tasks, "Peg Insert Side" and "Shelf Place"; these were therefore omitted in Table 4. Furthermore, we observe that utilizing a sparse reward signal only appears to have strong default performance, and achieves a higher overall average success rate than TADPoLe or VLM-RM. This showcases the inherent power of a sparse reward in solving Meta-World tasks. However, there are two additional takeaways; firstly, VLM-RM surprisingly achieves a substantial decrease in performance from the default sparse reward signal, highlighting TADPoLe as a more preferable dense text-conditioned reward provider. Secondly, TADPoLe is able to solve more overall tasks than other approaches. In particular, TADPoLe can solve the "Door Open" task, which is completely unable to be solved by VLM-RM or using a sparse reward only. We therefore highlight TADPoLe as a promising text-conditioned reward signal in replacing ground-truth hand-designed feedback, particularly as the text-to-image diffusion model utillized was pretrained in a general manner, without finetuning explicitly on Meta-World demonstrations.

Table A5: **Meta-World Task-Prompt Pairs**

| Task | Text Prompt |
|------|-------------|
| Door Open | opening a door |
| Door Close | closing a door |
| Drawer Open | opening a drawer |
| Drawer Close | closing a drawer |
| Window Open | opening a window |
| Window Close | closing a window |
| Coffee Push | pushing a white mug towards the coffee machine |
| Button Press | pressing a button |
| Soccer | pushing a soccer ball into the net |
| Peg Insert Side | inserting a peg into the slot |
| Shelf Place | placing an object on the shelf |
| Lever Pull | pulling a lever |

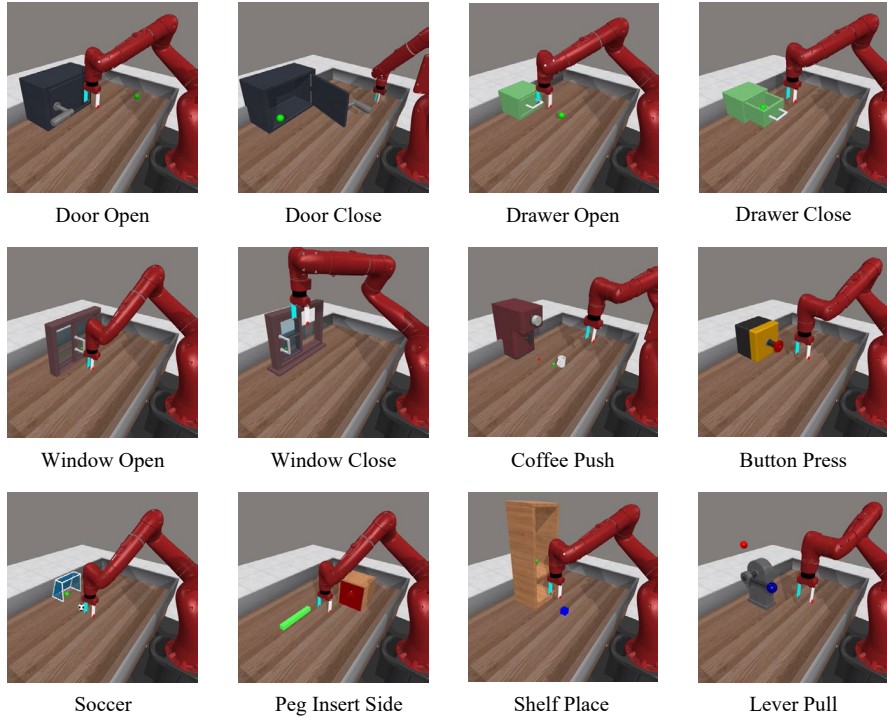

Figure A2: **Meta-World Tasks.** We select 12 robotic arm tasks from Meta-World suite as our evaluation task set, balanced in terms of diversity and complexity.

Table A6: Average success rate for 12 robotic manipulation tasks in Meta-World [42] over 30 evaluation rollouts. We compare between TADPoLe and VLM-RM, both approaches that do not utilize in-domain data or demonstrations, and find TADPoLe significantly outperforms VLM-RM. We report mean performance and standard deviation across 10 seeds for each task.

| Success Rate (%) | Door Open | Door Close | Drawer Open | Drawer Close | Window Open | Window Close | Coffee Push |
|---|---|---|---|---|---|---|---|
| Sparse | 0 (± 0) | 99.7 (± 1.0) | 74.7 (± 28.7) | 99.3 (± 2.0) | 94.3 (± 5.8) | 93.0 (± 16.9) | 46.3 (± 11.3) |
| VLM-RM | 0 (± 0) | 79.7 (± 39.8) | 10.0 (± 30.0) | 100 (± 0) | 9.7 (±29.0) | 0 (± 0) | 4.0 (± 12.0) |
| TADPoLe | 40.0 (± 49.0) | 100 (± 0) | 45.3 (± 46.6) | 100 (± 0) | 74.0 (± 37.9) | 30.0 (± 45.8) | 18.6 (± 20.5) |

| Success Rate (%) | Button Press | Soccer | Peg Insert Side | Shelf Place | Lever Pull | *Average* |
|---|---|---|---|---|---|---|
| Sparse | 86.7 (± 29.1) | 12.7 (± 14.1) | 0 (± 0) | 0 (± 0) | 0 (± 0) | 50.6 |
| VLM-RM | 30.0 (± 45.8) | 5.3 (± 11.6) | 0 (± 0) | 0 (± 0) | 11.3 (± 18.3) | 20.8 |
| TADPoLe | 73.0 (± 38.5) | 25.0 (± 15.4) | 0 (± 0) | 0 (± 0) | 0 (± 0) | 42.2 |

# D  Training Curves

In our experiments, we compare against other methods that provide text-conditioned rewards. As each of these methods are formulated differently, the difference in scales naturally prevent easy direct comparison. However, plotting the training curves for TADPoLe does yield insights into its speed of convergence, and investigating the visual performance at intermediate steps can be interesting. In Figure A3 we showcase the TADPoLe training curves for a variety of novel text-conditioned policies. We highlight the performance at steps 500k, 1M, 1.5M, and 2M in red; we further visualize the policy achieved at these intermediate steps by showcasing the last frame of the achieved video.

In the case of the policy learned for the prompt "a person standing with hands on hips", for example, we see that whereas the initial policies fail, the policy first learns to stand up by step 1.5M. Then, by the last training step, the policy has learned to correctly place its hands conspicuously on its hips. Similarly, for the prompt "a person kneeling", the policy first learns to kneel on all fours; by the end, the policy learns to successfully kneel on one knee.

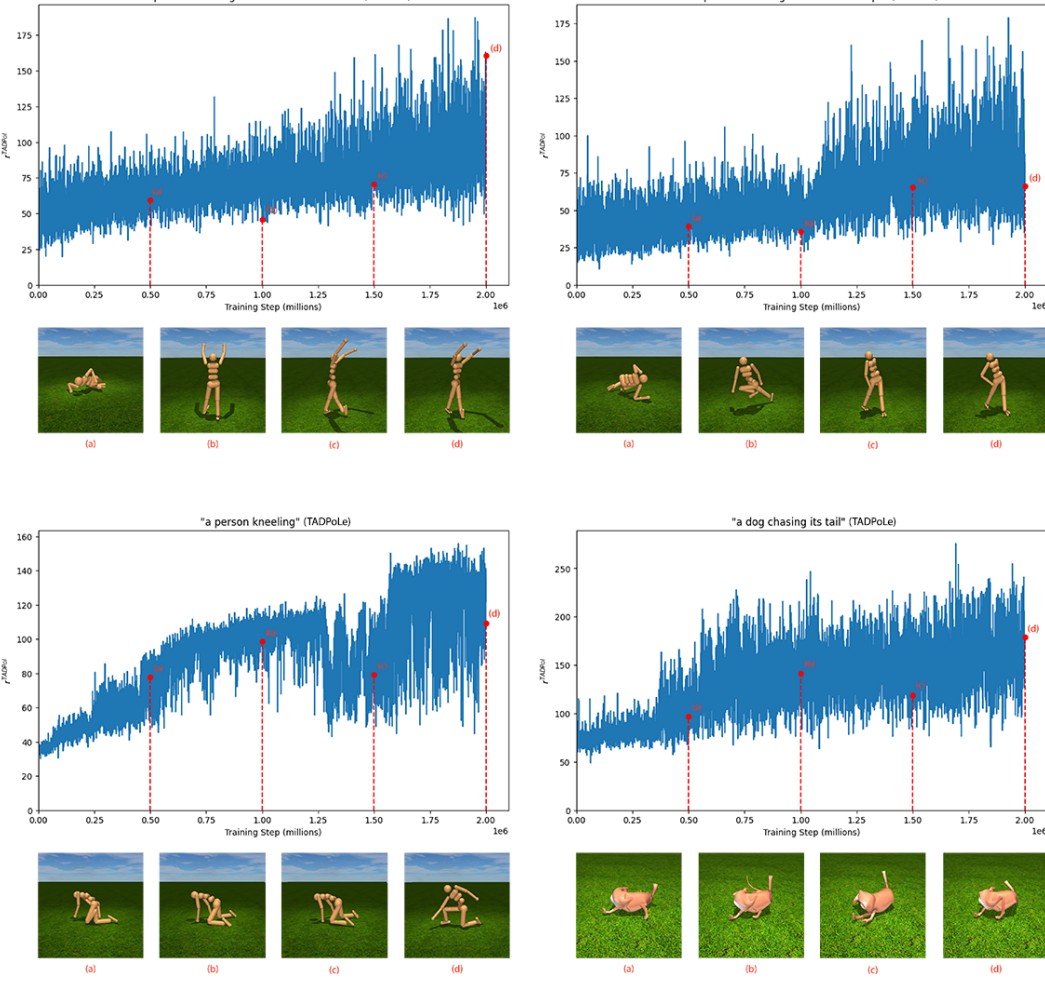

Figure A3: TADPoLe training curves for a variety of text-conditionings, with intermediate visualizations. The frames displayed are always the last frame achieved by the policy, at that particular training step.

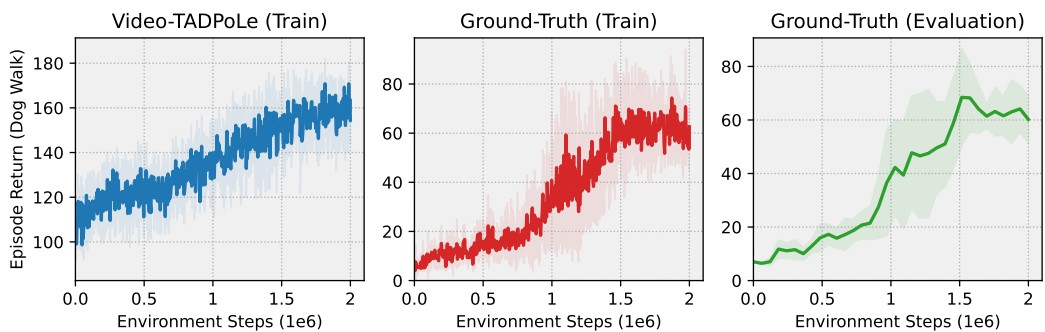

Figure A4: Episode return curves of a Dog Walk agent trained with Video-TADPoLe, using the prompt "a dog walking". From left to right: the cumulative Video-TADPoLe rewards achieved throughout training, the cumulative ground-truth rewards achieved throughout training, and the cumulative ground-truth rewards achieved during evaluation. Shaded regions denote the standard deviation across five random seeds.

Table A7: Numerical results from the user study on text-alignment. Out of 25 anonymous users, for each method, we report the percentage that believe the video achieved by a policy at the end of training is aligned with the provided text prompt. Entries above 50% are displayed in Table 1 as checkmarks (✔); below 50% are displayed as x-marks (✘).

| Environment | Prompt | VLM-RM | LIV | Text2Reward | TADPoLe (Ours) |
|---|---|---|---|---|---|
| Humanoid | "a person in lotus position" | 52% | 28% | 4% | **60%** |
| Humanoid | "a person doing splits" | 64% | **88%** | 0% | 84% |
| Humanoid | "a person kneeling" | 48% | 4% | 0% | **64%** |
| Dog | "a dog standing" | 16% | 0% | 40% | **84%** |

# E   Motivating Visualizations

For further intuition on TADPoLe's behavior, we visualize the complete denoising results for a query video achieved through a Dog policy and corrupted with some level of noise, conditioned on a consistent text prompt of "a dog walking". Note that this is purely for visualization purposes; in practice, when training policies, we do not visually denoise over multiple steps but instead extract a reward signal from components of a one-step denoising prediction. However, the multi-step denoising visualizations can offer us a glimpse of intuition into the behavior of the pretrained diffusion model and the properties of its predicted components at one single denoising step.

We first find that StableDiffusion, used in TADPoLe, is able to reconstruct a well-aligned policy rollout (a video of the Dog agent actually walking) relatively accurately frame-by-frame, for a noise level of 500 (Figure A5). However, the predictions can differ substantially for a misaligned policy rollout (a video of the Dog agent falling over rather than walking); for frames where the dog has collapsed on the floor, the StableDiffusion model still tries to respect the specified input prompt and attempts to predict dogs standing upright or walking, as depicted in Figure A6, resulting in a more noticeable difference between the achieved frame and predicted frame. This difference can be exploited to distinguish between well-aligned and misaligned policy rollouts, lending confidence to StableDiffusion as a supervisory signal for text-conditioned policy learning. For a higher noise level, StableDiffusion quickly forms hallucinatory final predictions (depicted in Figures A7, A8), but preserves more of the structure of the original video when the achieved video and provided text prompt is aligned.

We also visualize reconstruction predictions using a text-to-video model, namely an AnimateDiff v2 checkpoint, in Figures A9 and A10. We find that the motion prior is indeed meaningful, enabling it to reconstruct the Dog with a coherent texture and pose over time. In Figures A11, A12 we highlight how a text-to-video model is more robust in reconstructing the achieved video by the policy at high noise levels compared with StableDiffusion (Figures A7, A8). This supports the hypothesis outlined in Section A.1.

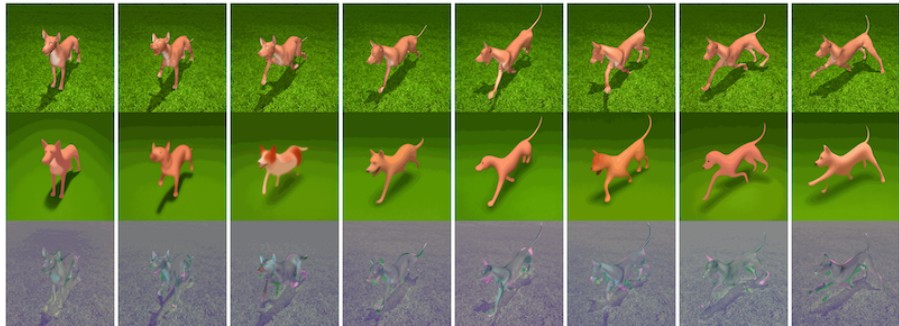

Figure A5: Visualizing the denoising of a good query trajectory from a noise level of 500 to completion using per-frame StableDiffusion.

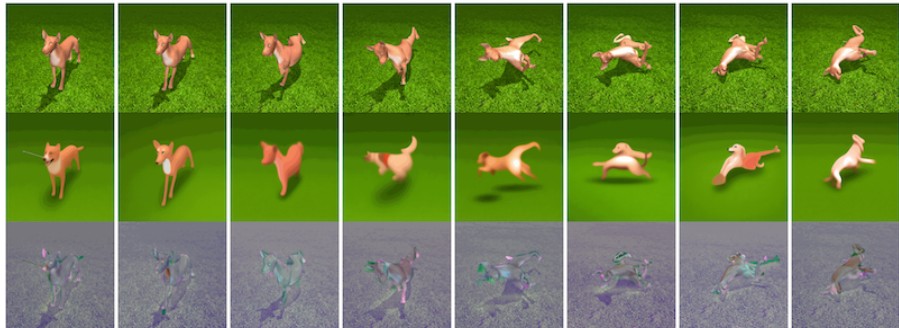

Figure A6: Visualizing the denoising of a failed query trajectory from a noise level of 500 to completion using per-frame StableDiffusion.

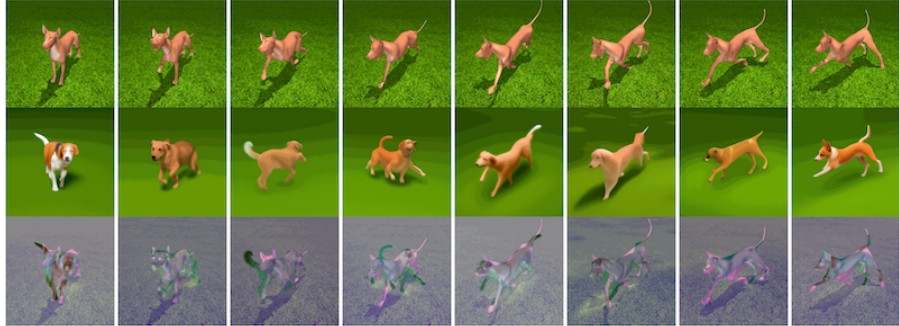

Figure A7: Visualizing the denoising of a good query trajectory from a noise level of 700 to completion using per-frame StableDiffusion.

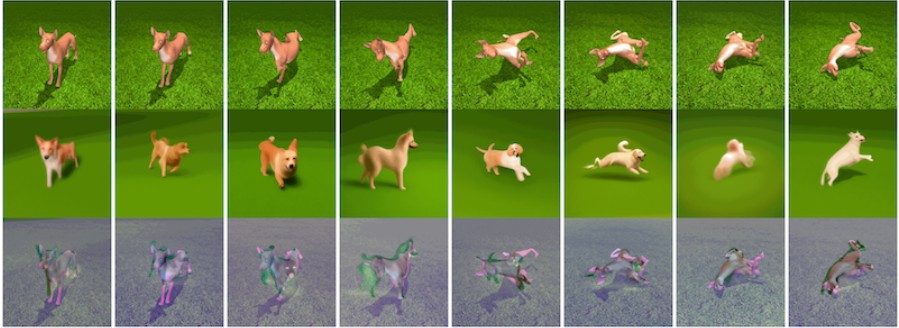

Figure A8: Visualizing the denoising of a failed query trajectory from a noise level of 700 to completion using per-frame StableDiffusion.

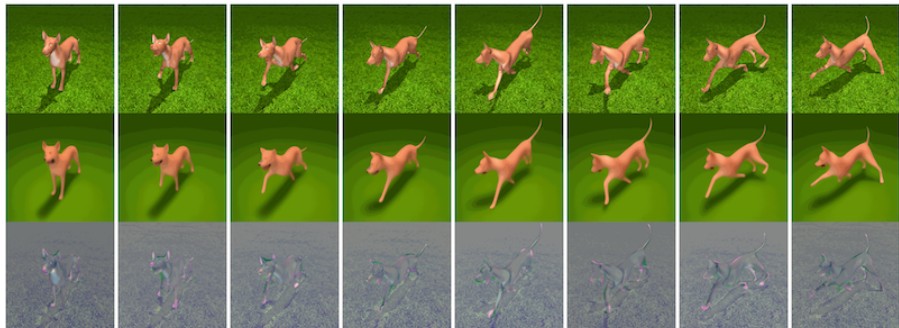

Figure A9: Visualizing the denoising of a good query trajectory from a noise level of 500 to completion using AnimateDiff.

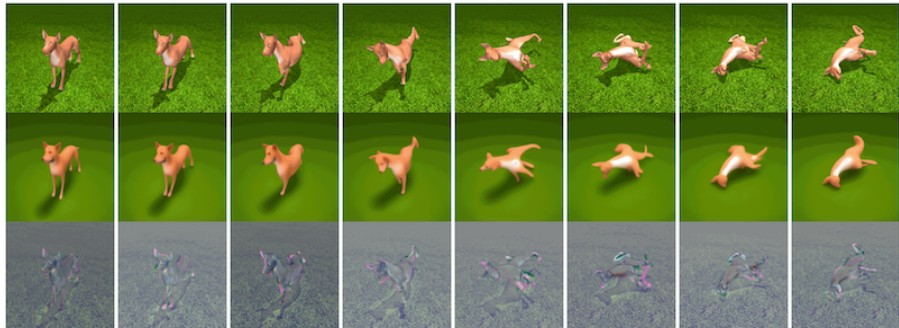

Figure A10: Visualizing the denoising of a failed query trajectory from a noise level of 500 to completion using AnimateDiff.

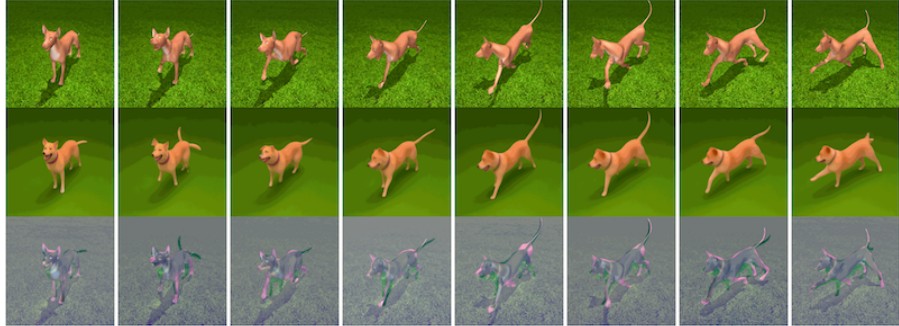

Figure A11: Visualizing the denoising of a good query trajectory from a noise level of 700 to completion using AnimateDiff.

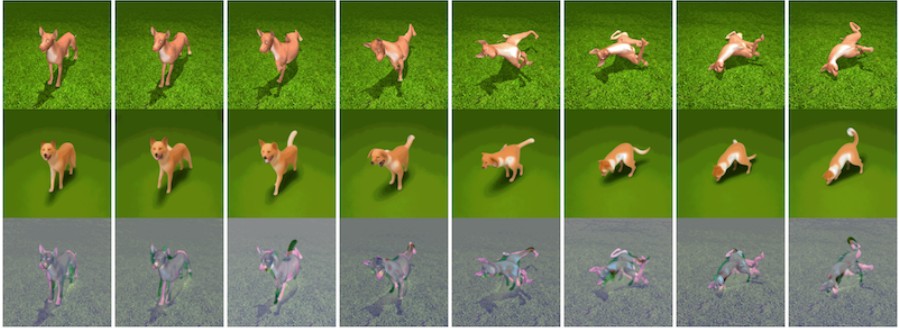

Figure A12: Visualizing the denoising of a failed query trajectory from a noise level of 700 to completion using AnimateDiff.

