# OpenReview forum: "Text-Aware Diffusion for Policy Learning"
_NeurIPS.cc/2024/Conference — NeurIPS 2024 poster_

### Official Review · Reviewer_KsL9 · 2024-06-12

**Soundness:** 4
**Presentation:** 4
**Contribution:** 4
**Rating:** 7
**Confidence:** 4

**Summary:**

The paper proposed a reward generating pipeline leveraging a text-conditioned diffusion model with a text prompt to master RL tasks described by the prompt.
The pipeline compares the difference of the generated image with/without the prompt and the original image to calculate a dense reward, which also makes sense intuitively.

**Strengths:**

- The authors cleverly use a conditional diffusion model to estimate the alignment level of an image to the text prompt with a hypothesis that  diffusion models capture naturalness and world knowledge.
- The authors expand the proposed method from image-text space to video-text space.
- As it is hard to quantitatively evaluate the performance of benchmarks, the paper conducted user studies to compare benchmarks by human volunteers.

**Weaknesses:**

- The proposed method requires more hyperparameter-tuning of the noise level and the overall reward function. Thus, more investigation is needed before training is conducted, such as Appendix A.
- The proposed method heavily relies on the capability of the external diffusion model to obtain world knowledge. It is a natural sin due to the complexity of the tasks the paper is solving.
- Another minor weakness is that some details of the user study are not presented, such as the variance of ✔ and ✘ and scores (instead of giving the majority and avg).

**Questions:**

- Is $\hat{\epsilon}_\phi$ a neural network (Sec 3.1) or a noise (Fig.3)?
- What is the cons of replacing the diffusion model in your method with a CLIP-like model to access the alignment level of image and text? One I can think of is the naturalness cannot be calculated in such case.
- Why is VLM-RM outperforming the proposed method in the standing Humanoid environment? Is it due to the training method of VLM-RM or because it is easier?
- Is it possible to give a concrete example of the failure mode mentioned in the limitation section?
- It would be interesting if the authors can try difficult tasks that even the diffusion model does not know how to generate and see the corresponding results of the RL training and evaluation.

**Limitations:**

See weakness 1. The paper concluded most of its limitation in the main text and Appendix.

---

> ### Author Rebuttal · Authors · 2024-08-07
>
> We thank Reviewer KsL9 for their thoughtful comments and helpful feedback on our work. Below, we seek to both address the reviewer’s listed weaknesses and answer the posed questions:
>
> **On hyperparameter tuning:** we concur with the reviewer that our approach benefits from initial tuning of the noise level, and reward computation terms such as $w_1$ and $w_2$. However, we demonstrate that such hyperparameter settings generally transfer across environments and robotic states with minimal modification. For example, the Humanoid, Dog, and MetaWorld experiments were conducted with the same TADPoLe noise level and $w_1, w_2$ terms. The same $w_1, w_2$ terms were used for Video-TADPoLe, and were also kept consistent between Humanoid and Dog environments - only the noise level range changed (with justification and discussion in Appendix A.1). Therefore, although there are more terms to tune, tuning does not generally need to occur ad hoc but can be preserved flexibly across environments with decent effect.
>
> **On relying on external diffusion models for world knowledge:** we agree with the reviewer’s analysis that the diffusion model is utilized to provide world knowledge to a policy during optimization. As demonstrated in the paper, this offers a natural way for priors summarized from large-scale pretraining to benefit policy learning, enabling the learning of policies that act and behave according to “natural” poses and motions summarized from natural image and video datasets, as well as enabling flexible text-conditioning. As diffusion models improve in encoding world knowledge, stemming from efforts on understanding better physics, dynamics, etc. we anticipate the quality of policies learned through TADPoLe to directly improve as well.
>
> **On user study details:** In Table A7 of our rebuttal Appendix page, we provide the raw user study results on text-alignment, which correspond to the checkmark and x-marks in Table 1. Among 25 anonymous and randomly selected online participants, we deem a prompt to be correctly learned when more than 50% of the participants vote for text-behavior alignment (denoted by a checkmark). We observe that TADPoLe generally has the highest positive vote ratios.
>
> **Clarifying $\mathbf{\hat\epsilon}_{\boldsymbol{\phi}}$:** we thank the reviewer for highlighting the notational discrepancy; indeed, we overload $\mathbf{\hat\epsilon}_{\boldsymbol{\phi}}$: as a standalone it represents a noise prediction, and when utilized as a function it represents the neural network that produces the prediction. We will clarify this overloaded notation in the caption of Figure 3 in the final draft of the paper.
>
> **On replacing the diffusion model with CLIP-style:** indeed, the reviewer’s suggestion is what VLM-RM is doing, which we compare against in our experiments. Utilizing a diffusion model over a CLIP-based approach has conceptual benefits aligned with the intuition of the reviewer. As a CLIP-based alignment model is trained on clean pairs of image and text, during inference it therefore assumes that the input for the image is always clean. During ordinary retrieval, from a set of natural images, this is not an issue. During policy optimization, however, it essentially has the flexibility to search for the most text-aligned image amongst potentially unnatural-looking poses. Having a base assumption that every query image is “natural”, it may therefore cause the policy to learn an unnatural-looking pose but the model deems highly-aligned with the text prompt. On the other hand, a diffusion model is trained to not only respect text-alignment, but also understand what is a natural-looking image at all; as a generative model, it seeks to generate a natural-looking image and therefore must learn a notion of "naturalness" from its pretraining data. Therefore, an approach like TADPoLe can result in more natural-looking policies learned (as evidenced by the qualitative results reported in Tables 1, 2, and 3 in comparison to VLM-RM).
>
> **On VLM-RM standing performance:** another distinguishing factor between using a CLIP-style model and a diffusion model in our method is that CLIP provides a deterministic reward signal. In other words, for the same query image and query text prompt pair, the CLIP model will always provide the same alignment reward score. When utilized for policy optimization, for a fixed text prompt, the policy can essentially be seen as taking actions to search for the singular frame that will achieve the highest deterministic alignment score and then maintaining it for perpetuity. Therefore, goal-achievement tasks like Humanoid Stand are very stable for CLIP-based approaches to learn; it learns to achieve a standing position and maintain it. However, there are two drawbacks: having a singular deterministic score which encourages goal-achievement does not translate well to continuous locomotion, where there is no canonical pose to achieve, and the pose that it does achieve may not be aligned with a notion of naturalness as mentioned above.
>
> **On failure modes:** In our updated submission website, we provide Video-TADPoLe results for the prompts “a person walking to the left” and “a person walking to the right”. Whereas there are seeds of the Humanoid successfully walking with respect to the provided prompt, there are also seeds where the person walks in the opposite direction. How to provide fine-grained control over the text-conditioning to focus on key words such as direction (beyond simply walking) is interesting to explore for future work; furthermore, a potential way to remedy this limitation is with improvements to the underlying text-conditioned diffusion model itself, such as with motion adapters.

---

> > ### Comment · Reviewer_KsL9 · 2024-08-09
> >
> > Thanks for the detailed rebuttal.

---

### Official Review · Reviewer_TWnR · 2024-07-03

**Soundness:** 3
**Presentation:** 1
**Contribution:** 2
**Rating:** 4
**Confidence:** 4

**Summary:**

The paper introduces Text-Aware Diffusion for Policy Learning (TADPoLe), a method for reinforcement learning that leverages pretrained text-conditioned diffusion models to compute dense reward signals. This approach allows agents to learn text-aligned behaviors without the need for expert demonstrations or handcrafted reward functions. The authors demonstrate the effectiveness of TADPoLe in various environments, including Humanoid, Dog, and Meta-World, achieving zero-shot policy learning with natural language inputs.

**Strengths:**

The paper provides a novel approach to reinforcement learning by using large-scale pretrained generative models to generate reward signals based on natural language descriptions. This method removes the need for manually crafted reward functions, which are often a bottleneck in RL tasks. The approach is well-motivated, leveraging the rich priors encoded in generative models trained on vast datasets. The experimental results show the ability of TADPoLe to learn diverse behaviors across different environments and tasks.

**Weaknesses:**

1. Presentation Issues: The paper's presentation is problematic. The integration of components such as the 'symlog' of the reward function and the 'noise level range' is not clearly explained. It is difficult to understand the necessity and utility of these components and how much performance improvement depends on them. Additionally, the paper lacks clarity on the source of ground-truth for rendered subsequent images and whether the method requires pre-existing well-rendered videos for each environment for TADPoLe training.

2. Alignment with Motivation: The motivation to leverage text-conditioned diffusion models for reward generation is clear; however, the demand for well-rendered videos for each environment does not fully align with the goal of making reinforcement learning more practical and scalable. Some environments donot have existing well-performed video for the agent, so will this method require for a pre-trained policy to collect the video data?

3. Experimental Insufficiency: The experiments are not comprehensive. There is a need to expand the range of tasks to provide a more thorough evaluation of the method. Additionally, the paper had better include comparisons with the Diffusion-Reward method [1] to highlight the advantages of TADPoLe.

4. The ablation study is also insufficient. It should further explore the design of the 'symlog' component and the selection of the two weights, as these operations lack detailed justification. This analysis is crucial to understand the contribution of each part to the overall performance and to provide a clearer justification for their inclusion.

5. Overclaim. The writing of the paper is dense and lacks clarity in several key areas. The explanations of the methodology and components are not sufficiently detailed, making it challenging for readers to grasp the full picture of how TADPoLe functions and why certain design choices were made. Improving the clarity and coherence of the writing would significantly enhance the paper's readability and accessibility.

[1] Huang, T., Jiang, G., Ze, Y., & Xu, H. (2023). Diffusion Reward: Learning Rewards via Conditional Video Diffusion. arXiv preprint arXiv:2312.14134.

**Questions:**

1. Clarification of Component Integration: Why is the 'symlog' transformation necessary? How does it and 'noise level range' specifically contribute to the performance improvements observed? More detailed explanations and justifications are needed.

2. Source of Ground-Truth: Where do the ground-truth rendered subsequent images come from? Does the method rely on pre-existing well-rendered videos for each environment? This aspect should be clarified to understand the method's applicability.

3. Experimental Expansion: To provide a more comprehensive evaluation, expand the range of tasks and include comparisons with the Diffusion-Reward method. This will help to position TADPoLe's performance in a broader context.

4. Ablation Study: Conduct a more detailed ablation study to isolate the effects of the 'symlog' transformation and the specific weightings used. This will help to understand the contribution of each part to the overall performance and provide a clearer justification for their inclusion.

5. Scalability and Practicality: Address the scalability issue regarding the need for well-rendered videos in environments lacking existing well-performed videos. How does the method handle such cases, and what are the implications for its practicality and scalability?

**Limitations:**

The author has addressed a limitation. It's also recommended to add the unsolvable questions to the limitations.

---

> ### Author Rebuttal · Authors · 2024-08-07
>
> We thank Reviewer TWnR for their detailed comments and thorough questions. We seek to address their concerns below:
>
> **On pre-existing well-rendered videos:** we would like to clarify a potential misunderstanding - our method does not require any pre-existing well-rendered videos for environments TADPoLe is applied to, and therefore there is no source of ground-truth image/video demonstrations for arbitrary environments (no offline videos or demonstrations are collected, and pre-existing expert policies are not needed). Rather, we utilize the rendering **function** at each timestep **on-the-fly** to generate a dense text-conditioned reward, as computed by a large-scale generally-pretrained diffusion model. We do not update the diffusion model with in-domain examples whatsoever. We therefore showcase how large-scale pretraining over natural images, videos, and text-alignment can directly transfer to the *zero-shot optimization* of policies that behave in alignment with text-conditioning, as well as natural-looking goal poses or motions.
>
> **On alignment with motivation:** we appreciate that the reviewer believes our motivation on using text-conditioned diffusion models for reward generation is clear; in light of our clarification that TADPoLe can be applied to arbitrary environments without demanding well-rendered video examples a priori (and therefore no pre-trained policies are needed either), we believe that our final proposed approach is well-aligned with the goal of making reinforcement learning more practical and scalable. Indeed, TADPoLe offers an avenue for supervising policies that behave not only in alignment with natural language, but also in alignment with natural-looking poses or motions captured from large-scale pretraining on natural images and videos.
>
> **On ablations:** the reviewer has highlighted three design decisions of interest: the *symlog* operation, the *noise level range*, and the selection of the two weights $w_1$ and $w_2$. We utilize the *symlog* operation to normalize the scales of the computed reward (in conjunction with $w_1$ and $w_2$ terms); indeed, we observe in Figure A1 where over all noise levels, the final computed reward stays roughly between 0 and 1, despite changes to the visual input or text prompt. Furthermore, this consistent normalization enables the reuse of hyperparameter settings (such as noise level, weights $w_1$ and $w_2$) across tasks and environments and even other diffusion models (e.g. Video Diffusion models) for consistent policy learning. We demonstrate this in a quantitative manner in Table A6 of the updated rebuttal Appendix page, by showcasing how removing the symlog normalization reduces consistent policy learning across environments and diffusion models with the same hyperparameter settings. For the *noise level range* used, we have provided much justification in the Appendix of our original paper submission through a discussion (Section A, A.1) as well as graphical Figures (Figure A.1). Similarly, in our original submission we have also reported that the values of the two weights $w_1$ and $w_2$ were selected through a hyperparameter sweep on Humanoid standing and walking performance (Section 4.1, Implementation), which we provide in Table A4, along with a discussion on their effects in Appendix C.3.
>
> **On experimentation:** the focus of our work is evaluating zero-shot optimization of policies conditioned on natural language. We highlight that text-conditioned policy learning through foundation models is a recent and active area of exploration. We compare against other recent work in VLM-RM (ICLR 24), LIV (ICML 23), and Text2Reward (ICLR 24), while also proposing novel baselines (ViCLIP-RM). We further highlight the diversity of our baselines to position TADPoLe in a broader context: Text2Reward uses a language-only approach to create a text-conditioned reward function, whereas VLM-RM, LIV, and ViCLIP-RM compute rewards on-the-fly in a multimodal manner. Furthermore, our range of tasks includes not only Dog and Humanoid (Table 1, 3), but also robotic manipulation tasks from MetaWorld (Table 4). We extend our range of tasks to include multiple novel prompts, as well as those with subtle details (Figure 4).
>
> **On comparison with Diffusion-Reward:** our work focuses on learning text-conditioned policies in a zero-shot manner through a large-scale generally-pretrained diffusion model. Diffusion-Reward (concurrent work, to be published at ECCV 24) is not a natural comparison for our approach, as it does not enable text-conditioned policy learning (their diffusion model is conditioned only on a history of frames), and their method explicitly requires expert video demonstrations from the environment to train their in-domain diffusion model. On the other hand, TADPoLe (along with listed baselines like VLM-RM, LIV, Text2Reward) requires no in-domain or expert demonstrations, and is text-conditioned. However, we have adapted a version of Diffusion-Reward for comparison purposes, where rewards are conditioned not on history but on a natural language input, and reported it in Table A8 of the rebuttal Appendix page. We demonstrate that TADPoLe outperforms the adapted Diffusion-Reward implementation in terms of overall success rate aggregated over all tasks in the suite. We further note that Diffusion-Reward requires multiple denoising steps for each dense reward computation (in practice they use 10 steps), whereas TADPoLe only requires one denoising step per reward computation, thus substantially improving in computation speed and complexity.

---

> > ### Comment · Reviewer_TWnR · 2024-08-09
> > **Response to rebuttal**
> >
> > Dear Authors,
> >
> > Thank you for your detailed rebuttal and the clarifications provided. I appreciate your effort in addressing my concerns, especially the clarification that there is no need for expert data or pre-existing well-rendered videos. This has resolved a significant part of my concerns, and I am inclined to adjust my rating to 4 in recognition of it.
> >
> > However, I still have some remaining concerns that I believe are important to address:
> >
> > Symlog Regularization: The additional results in Table A6 from your rebuttal indeed highlight the significance of the symlog normalization. This brings up a question regarding the flexibility and robustness of your method—have you considered or experimented with alternative regularization methods? It would be insightful to see if other techniques could provide comparable or even improved performance, which would further strengthen the generalizability of your approach.
> >
> > Experimental Rigor: While I appreciate the addition of Table A8, which compares your method with an adapted version of Diffusion-Reward, I agree with the general sentiment among reviewers that the experimental evaluation remains somewhat weak. The results from MetaWorld, where most tasks either achieve 100% or 0% success rates, suggest a potential limitation in the diversity and complexity of the tasks. Incorporating more challenging environments, such as those involving interactive tasks with Franka-Kitchen or the dexterous manipulation tasks with Adroit, would provide a more convincing demonstration of your method's capabilities. This would be crucial for further validating the practicality and robustness of TADPoLe.
> >
> > Presentation: Finally, I recommend revisiting the presentation of your paper. Specifically, Figure 3 could be redesigned to locate in a line with Figure 2, allowing for space to provide a more detailed and visually appealing presentation of your experimental outcomes.
> >
> > Best regards.

---

> > > ### Author Response · Authors · 2024-08-14
> > > **Response to Reviewer [1]**
> > >
> > > We thank the reviewer for the response; we are happy to have addressed many of the prior concerns, and appreciate the increase in score. We would like to supply additional experimental results towards the interest of the reviewer:
> > >
> > > **On alternative normalization:** Indeed, we have previously tried alternative normalization techniques for our reward computation. Apart from symlog and using simple scaling factors directly, we have also tried using symexp, min-max rescaling (subtract an empirical min, divide by an empirical min-max range, and rescaling to [-1, 1]), as well as standardization (subtracting an empirical mean and dividing by an empirical standard deviation). What we have found is that the symlog operation overall is the best reward normalization strategy, in terms of transferring hyperparameter configurations across robotic configurations, visual environments, and desired tasks.  We further note that min-max rescaling and standardization require empirically estimated values, whereas symlog does not.  We provide an updated table:
> > >
> > > |Task|Prompt|TADPoLe|TADPoLe-simple-scale|TADPoLe-symexp|TADPoLe-min-max|TADPoLe-standardization|Video-TADPoLe|Video-TADPoLe-simple-scale|Video-TADPoLe-min-max|Video-TADPoLe-symexp|Video-TADPoLe-standardization|
> > > |:-|:-:|:-:|:-:|:-:|:-:|:-:|:-:|:-:|:-:|:-:|-:|
> > > |Humanoid-Stand|“a person standing”|267.23|**276.67**|241.49|256.81|236.59|-|-|-|-|-|
> > > |Humanoid-Walk|“a person walking”|-|-|-|-|-|**226.29**|4.58|3.68|61.31|134.66|
> > > |Dog-Walk|“a dog walking”|-|-|-|-|-|**81.22**|35.30|9.46|6.15|5.05|
> > >
> > > **On Figure 3:** we appreciate the reviewer’s presentation suggestions, and agree that moving Figure 3 to the same row as the ones in Figure 2 could allow additional space.  We have already provided experimental outcomes of goal-achieving policies for Figure 1 and Figure 4, and have left continuous locomotion demonstrations for our attached videos on our website.  However, the additional space from moving Figure 3 could be used to explicitly show static frame rollouts corresponding to the videos on the website within the PDF.

---

> > > > ### Author Response · Authors · 2024-08-14
> > > > **Response to Reviewer [2]**
> > > >
> > > > On additional robotic experiments: In our experimental sweeps, we have been able to get TADPoLe working on three Adroit tasks (door, pen, hammer), in a similar setup with what is reported in MetaWorld (all hyperparameters are kept the same, e.g. noise level, $w_1$, $w_2$, etc.).  For the base RL backbone algorithm, we find the most success on Adroit from applying the text-conditioned TADPoLe reward to DrM [1].  The associated task prompts are “opening a door” and “spinning a pen”, and “hammering a nail” respectively, and we include the success rate and video demonstration on the bottom of our updated [submission website](https://sites.google.com/view/tadpols/home).  We hope these positive results on complex dexterous manipulation tasks provide a more convincing demonstration of TADPoLe’s capabilities to the reviewer.
> > > >
> > > > We also voluntarily report, for complete transparency, that of the FrankaKitchen suite, in which we select five tasks for evaluation following R3M [2], our approach is only able to solve one of the five tasks (turning the light on) when using the same hyperparameter settings as the MetaWorld experiments.  In fact, LIV [3], in Appendix G4, explicitly states that **fine-tuning is needed for language-grounding** for FrankaKitchen tasks (compare Figure 20 against Figure 21), to form a coherent language-goal reward signal for text-conditioned policy learning. In light of this, TADPoLe performance on FrankaKitchen is expected to be difficult, as TADPoLe uses no fine-tuning at all, but still tries to provide a text-conditioned reward.  The performance TADPoLe can achieve on robotic environments (such as Adroit, MetaWorld, and the FrankaKitchen example) while utilizing *no in-domain demonstrations, or fine-tuning whatsoever* is promising, particularly as such robotic experiments generally have a large visual domain gap from the pretraining data observed by the general-purpose diffusion model (and collecting in-domain examples to alleviate the domain gap can be expensive).
> > > >
> > > > We believe our framework is general enough that future improvements to the underlying diffusion model, for example finetuning on in-domain demonstrations, would help - but our focus in this work is on the *inherent generalization capabilities* from *leveraging large-scale pretrained diffusion models*, which we demonstrate is surprisingly powerful.  As our framework still works with arbitrary image and video diffusion models, we leave exploring the utilization of diffusion models that have explicitly integrated in-domain information (such as through explicit finetuning) for exciting future work.
> > > >
> > > > The main focus of our work is finding a general-purpose, scalable solution for text-conditioned policy learning - rather than solving the entire realm of text-conditioned policy learning for robotics at once, or outperforming even models that leverage in-domain demonstrations.  What we seek to show is that text prompts can be converted into dense rewards without in-domain finetuning, just from text-conditioned visual diffusion models pretrained on general data, to enable policy learning.  Our model formulation, as well as our reward computation are novel, and we demonstrate this generalizes across a variety of robotic environments (Humanoid, Dog, MetaWorld), and across a variety of text prompts and novel behaviors without *finetuning or modification*, and with large reuse in hyperparameters.  Our approach seeks to be general, and scalable - motivations that the reviewer has voiced appreciation for, and accomplishes this by avoiding any reliance on in-domain information - but this generality often comes at the cost of achieving state-of-the-art performance on every single domain, which we hope is understandable without detracting from the surprising insights TADPoLe already provides.
> > > >
> > > > We appreciate the reviewer’s suggestions for strengthening our work, and plan to incorporate these updates into the final camera-ready version of the paper.
> > > >
> > > > [1] Xu et al. DrM: Mastering Visual Reinforcement Learning through Dormant Ratio Minimization. ICLR, 2024.
> > > >
> > > > [2] Nair et al. R3M: A Universal Visual Representation for Robot Manipulation. arxiv preprint, 2022.
> > > >
> > > > [3] Ma et al. LIV: Language-Image Representations and Rewards for Robotic Control. ICML, 2023.

---

### Official Review · Reviewer_iiYV · 2024-07-15

**Soundness:** 2
**Presentation:** 3
**Contribution:** 3
**Rating:** 6
**Confidence:** 3

**Summary:**

The paper presents Text-Aware Diffusion for Policy Learning (TADPoLe), a framework that leverages pretrained text-conditioned diffusion models to generate dense, zero-shot reward signals for policy learning in reinforcement learning tasks. The approach aims to address the limitations of manually designed reward functions by utilizing large-scale generative models to encode rich priors that guide policies in a natural and text-aligned manner. Experiments demonstrate TADPoLe’s effectiveness in learning policies for novel goals and continuous locomotion behaviors in various environments, including humanoid, dog, and robotic manipulation tasks, without ground-truth rewards or expert demonstrations.

**Strengths:**

**Strengths:**

- **Innovative Reward Generation:** TADPoLe introduces a novel approach to reward signal generation using pretrained diffusion models, reducing the need for manually crafted reward functions.
- **Zero-Shot Learning:** The framework supports zero-shot policy learning, enabling the agent to learn new tasks and behaviors from natural language descriptions without prior demonstrations.
- **Diverse Applications:** Demonstrates versatility across different environments and tasks, including humanoid and dog locomotion, and robotic manipulation in the Meta-World environment.
- **Human Evaluation:** Qualitative assessments show that the policies learned by TADPoLe are perceived as more natural and aligned with the provided text prompts by human evaluators.

**Weaknesses:**

**Weaknesses:**

- **Evaluation Metrics:** While the paper provides qualitative evaluations, the reliance on human judgment for assessing the naturalness and alignment of behaviors could introduce subjectivity.
- **Scalability of Text Prompts:** The approach may struggle with very complex or ambiguous text prompts, and the extent to which it can handle highly detailed or context-specific instructions is not fully explored.
- **Comparative Baselines:** The comparisons with other text-to-reward methods are limited, and it would be beneficial to include more diverse baselines to better understand the relative performance of TADPoLe.
- **Computational Overhead:** The approach involves significant computational overhead due to the use of large-scale diffusion models, which could limit its practicality in resource-constrained environments.

**Questions:**

### Questions

1. **Reward Scaling:** How does TADPoLe handle the scaling of reward signals across different tasks and environments to ensure consistent policy learning?
2. **Text Prompt Complexity:** Can the method be extended to handle more complex and detailed text prompts? What are the limitations in terms of prompt complexity and specificity?
3. **Long-Term Dependencies:** How does TADPoLe perform in tasks requiring long-term planning and dependencies? Are there any mechanisms to ensure temporal coherence in learned behaviors?
4. **Generalization:** How well does the approach generalize to unseen environments or tasks that are significantly different from the training scenarios?

**Limitations:**

Same as the above section.

---

> ### Author Rebuttal · Authors · 2024-08-07
>
> We thank Reviewer iiYV for their helpful feedback and suggestions. We try to address their concerns below:
>
> **Qualitative evaluation:** We report quantitative comparisons whenever available (Tables 1, 3, 4), and we agree that human evaluation may inevitably introduce subjectivity. However, they are necessary when mere quantitative metrics are not sufficient to capture the performance difference, for example when evaluating the novel text-conditioned behaviors. We attempt to make such evaluation as impartial as possible; we use the videos of policy behavior at the last timestep of training without cherry-picking, and query 25 random participants without prior exposure to the task through an anonymized platform to estimate a general response from the human population. We provide details of our qualitative evaluation procedure in Section 4.1, and also additional fine-grained user study results in our updated rebuttal Appendix page (Table A7).
>
> **On text prompt scalability and complexity:** we agree that exploring text prompt scalability and sensitivity is important. We refer the reviewer to Figure (4), where we report preliminary investigations on how sensitive the method is when text prompts are extended with details (“a person standing” -> “a person standing with hands above head”), and when subtle details are modified in the text conditioning (“a person standing with hands above head” -> “a person standing with hands on hips”). However, despite showing that TADPoLe indeed demonstrates a level of sensitivity and scalability to long, detailed, text prompts, we agree that extremely complex prompts, such as multiple sequential instructions over time, may still be a challenge - developing this is a worthwhile future direction. We also note that our method is generally applicable to both image and video generative models, and therefore provides a way to handle text prompts based on both visual appearance as well as motion. As the modeling power of base text-to-video diffusion models improves, so too do we expect our approach to scale to respect more complex text prompts accurately.
>
> **On comparative baselines:** we highlight that text-conditioned policy learning through foundation models is a recent and active area of exploration. We compare against other recent work on text-aware rewards in VLM-RM (ICLR 24), LIV (ICML 23), and Text2Reward (ICLR 24), while also proposing novel baselines (ViCLIP-RM). We further highlight the diversity of our baselines: Text2Reward uses a language-only approach to create a text-conditioned reward function, whereas VLM-RM, LIV, and ViCLIP-RM compute rewards on-the-fly in a multimodal manner. In addition, in the new Table A8 of our rebuttal Appendix page, we further compare with LIV and Diffusion-Reward (a concurrent work, to be published at ECCV 24) on Meta-World, where TADPoLe enjoys the best performance.
>
> **On computational overhead:** Our reward computation only uses one denoising step to generate text-conditioned rewards, which avoids the overhead of multiple iterations of denoising steps usually performed with generating data through diffusion models in vanilla inference. Furthermore, our reward computation is general across diffusion model implementations, and the complexity of the diffusion model can be adjusted (such as through distillation) to fit desired resource constraints while still utilizing our proposed reward computation.
>
> **On reward scaling:** we discover that the symlog operation helps to normalize the raw computed reward signals across tasks and environments to be roughly on the same scale. We are therefore able to reuse hyperparameter settings (such as noise level, weights $w_1$ and $w_2$) across tasks and environments and even other diffusion models (e.g. Video Diffusion models) for consistent policy learning. We demonstrate this in a quantitative manner in Table A6 of the updated rebuttal Appendix page, by showcasing how removing the symlog normalization reduces consistent policy learning across environments and diffusion models with the same hyperparameter settings.
>
> **On long-term dependencies:** exploring how TADPoLe performs in tasks requiring long-term planning and dependencies is an interesting and worthwhile direction for future work. Due to the flexibility of text-conditioning, describing a task with long-term dependencies can be done in a way that is more like providing a sparse reward (e.g. describing just the desired goal) versus a manner that is more akin to providing a dense reward through multiple subtasks (e.g. providing detailed instructions). We anticipate the following considerations to potentially help solve long-term dependencies through TADPoLe: choosing appropriate underlying learning techniques (e.g. one that performs exploration, which can discover how to solve tasks that requires long-term dependencies), performing prompt tuning to discover the best prompts for a task, and learning to attend on different portions of a detailed text prompt (e.g. instructions) based off of progress of the policy’s performance.
>
> **On task generalization:** we would like to clarify that from the perspective of TADPoLe, all environments are unseen and novel; we utilize generally-pretrained diffusion model checkpoints, with no tasks or even examples of the downstream visual environment observed during pretraining. Policy supervision is computed purely from priors captured within diffusion models with large-scale pretraining and no in-domain examples are used to update the diffusion model. We believe that applying TADPoLe across Humanoid, Dog, and MetaWorld already demonstrates strong task generalization capability, as the agents and visual characteristics of each environment differ greatly from each other. Furthermore, we demonstrate how text-conditioned policies can be learned across these distinct environments without minimal modification to its hyperparameters, even generalizing across which pretrained diffusion model is used.

---

### Official Review · Reviewer_z3di · 2024-07-18

**Soundness:** 2
**Presentation:** 3
**Contribution:** 2
**Rating:** 4
**Confidence:** 4

**Summary:**

The paper introduces Text-Aware Diffusion for Policy Learning (TADPoLe), which uses a large-scale pretrained text-conditioned diffusion model to provide zero-shot reward signals for training agents without expert demonstrations or manually designed reward functions. TADPoLe enables agents to learn behaviors and achieve goals specified by natural language in various tasks, demonstrating its effectiveness.

**Strengths:**

1. The idea is straightforward and easy to follow; the authors have adopted a clear approach to present their insights.

2. Using text-conditioned diffusion models to provide rewards is novel.

**Weaknesses:**

1. Although the text-aware diffusion reward is novel, both text-aware rewards and diffusion rewards have been proposed by prior works. The paper lacks an apple-to-apple comparison, making it difficult to discern the specific advantages of using diffusion models.

2. I find the experiments to be quite limited. For example, the authors only use TD-MPC as the algorithm backbone, which is just one model-based RL algorithm and does not have a significant advantage in visual RL. Additionally, the curve shown in Figure 5 is quite odd—why not place the baseline and proposed method on the same graph and include curves for other prompts? Furthermore, the comparisons in the Metaworld experiments are minimal, making the experimental results less convincing.

3. The absence of real-world experiments makes it hard to assess the paper's contribution to the community.

4. The authors use a diffusion model to generate rewards at each step, but there is no detailed analysis of the computational cost and its impact on speed.

**Questions:**

1. Even with sparse rewards in MetaWorld, the state-of-the-art results are better than those presented by the authors.
2. Is the naturalness in Table 2 determined through user surveys? I believe that videos would provide more compelling evidence.
3. How do different lengths and forms of prompts affect the rewards? What impact would slight adjustments to the prompts during training have?

**Limitations:**

The authors discuss limitations in the paper about how to control the weight of each individual word in the prompt. However, I believe that the primary limitations lie in the efficiency of the diffusion model. This paper has many aspects that require further discussion and improvement.

---

> ### Author Rebuttal · Authors · 2024-08-07
>
> We thank Reviewer z3di for their comments and feedback on our work; we are happy to hear that the reviewer appreciates the novelty of our approach, and we seek to address their concerns below.
>
> **On comparisons:** we performed *apple-to-apple* comparisons with three recent text-aware rewards, namely VLM-RM (ICLR 24), LIV (ICML 23), and Text2Reward (ICLR 24). We utilize the exact same underlying model architecture, hyperparameters and optimization procedure, where only the text-aware reward is changed across methods, to make it as comparable as possible (Table 1, 3). We highlight that using diffusion models has superior or comparable quantitative performances against other text-aware rewards under an apple-to-apple comparison, while having superior text-alignment and naturalness benefits (Table 2). To the best of our knowledge, there is not a diffusion reward baseline which we can compare with in an apple-to-apple setup. A concurrent work (to be published at ECCV 24), Diffusion-Reward, is the only work that leverages diffusion models for reward modeling. However, it does not condition on natural language, and thus does not enable text-conditioned policy learning, which is a central focus of our work. Furthermore, Diffusion-Reward relies on in-domain expert video demonstrations, while TADPoLe does not.
>
> **On the use of TD-MPC:** we utilize TD-MPC as a standardized backbone to evaluate different text-aware reward methods in a comparable setting, to focus on text-conditioned reward quality. We choose TD-MPC due to it being the first model able to solve the complex Dog environment (given ground-truth rewards for walking and standing), with strong performance on Humanoid as well. Given its powerful modeling capability, we select it as a testbed for text-aware reward comparison.
> On Figure 5: we place the baseline reward provided by the ground truth function and the TADPoLe reward on separate plots because they have different scales and are not directly comparable. However, we show that side-by-side, the trends across the two rewards are positively correlated, which is desirable. For TADPoLe reward plots across novel prompts, which do not have associated ground truth reward functions, additional examples can be found in Figure A11 of the Appendix.
>
> **On Metaworld comparisons:** Most prior methods which report performance on MetaWorld rely on (often expert-produced) video demonstrations from a similar domain or the target environment directly. We therefore focus our comparison on other methods that utilize a large-scale pretrained model for zero-shot (no in-domain demonstrations) text-conditioned policy learning, namely VLM-RM. Other work (namely Diffusion-Reward) utilize in-domain diffusion models trained explicitly on expert data and do not explore text-conditioned policy learning, which is the focus of our work. Furthermore, most performant state-of-the-art methods on MetaWorld were trained directly on dense ground-truth reward functions, whereas we evaluate the ability to recreate such behavior from dense text-conditioned rewards in a zero-shot manner. For further comparison, however, we include additional LIV results, as well as results from an adapted version of the Diffusion-Reward reward computation that is conditioned on natural language rather than historical frames, and uses a general text-to-video diffusion model in Table A8 of the rebuttal Appendix page. TADPoLe maintains superior performance in aggregate across the task suite.
>
> **On real-world experiments:** we agree that real-world experimentation would be nice to have. However, our focus is on exploring text-to-policy capabilities through diffusion models, which is a novel approach. We demonstrate our approach not only on the difficult Humanoid and Dog environments, which have high action spaces and complex transition dynamics, but also on Metaworld. In reference, prior work (VLM-RM) that used vision-language model supervision also did not apply it to real-world environments. However, in terms of future work, we do agree with the reviewer that deploying on real-world robotics is an exciting direction for further exploration.
> On Computational Cost and Speed: the computational cost of TADPoLe is simply one denoising forward step of a pretrained diffusion model to generate each dense reward. In comparison, Diffusion-Reward requires multiple denoising steps to generate one singular reward (in practice, 10 steps are used). TADPoLe is therefore computationally cheaper and faster, when the same diffusion model is utilized, in terms of reward computation.
>
> **On naturalness via user surveys:** Evaluations on naturalness were performed through user surveys (25 random participants), using videos achieved by policies at the last step of training. These are provided on the [submission website](https://sites.google.com/view/tadpols/home), for reviewers to visualize.
>
> **On the effect of prompts on rewards:** We refer the reviewer to Figure (4), where we find that different lengths and slight adjustments to the prompt during training indeed impact the final performance of the policy. To begin with, we extend a prompt in length from “a person standing” to “a person standing with hands above head”, and observe that the longer description can be respected. Furthermore, when we perform slight adjustments to the prompt, with “a person standing with hands on hips”, we confirm that the policy learned is able to respect subtle details such as placement of the arms. We anticipate extensive prompt tuning to produce further benefits; but for the scope of our work (demonstrating a rich text-conditioned reward from large-scale pretrained diffusion models) we leave tuning the most performant prompt for a specific desired task to the enthusiast.

---

> > ### Comment · Reviewer_z3di · 2024-08-10
> >
> > The reviewer did not respond to my Question 1.
> > The explanation of TD-MPC is not sufficient, and no other algorithm backbone is used to prove the effectiveness of the method during the rebuttal period.
> > As for the reason for real-world experiments, diffusion-reward actually added a lot of real-world experiments. Without real-world experiments, I think the contribution to the community is very limited. So I decide to maintain my score.

---

> > > ### Author Response · Authors · 2024-08-11
> > > **Response to Reviewer**
> > >
> > > We appreciate Reviewer z3di’s prompt response and engagement; we offer further clarification and results on the provided comments:
> > >
> > > **On Question 1:** To the best of our knowledge, prior work on MetaWorld mainly use dense ground-truth rewards, in-domain expert demonstrations, or both (e.g. Diffusion-Reward, referenced by the reviewer, uses a combination of in-domain expert demonstrations and sparse rewards). A more comparable setup is their reported performance on “unseen” tasks (Table 7) where they still use in-domain expert demonstrations (which TADPoLe does not need) to train their video model but the tasks are novel. Among the two overlapping tasks (Drawer-Open and Window-Open), both TADPoLe and Diffusion-Reward achieve near perfect performance, outperforming recently published baselines, such as VIPER (which we note also relies on in-domain expert demonstrations). We are unaware of published work that achieves "state-of-the-art" performance under a comparable setup as TADPoLe (using neither in-domain expert demonstrations nor ground-truth dense rewards) - we welcome clarification on the work the reviewer is referring to.
> > >
> > > **Beyond TD-MPC:** As explained in our initial rebuttal, our main motivation is comparing different text-to-reward methods on a consistent, powerful, backbone for an apples-to-apples comparison (with hyperparameters, optimization, architectures, etc. fixed). We select TD-MPC for its competitive performance on the complex Dog and Humanoid tasks, which serve as ideal environments to demonstrate **further** flexible novel behavior synthesis conditioned on text and naturalness. For example, for traditional model-free approaches that do not even have basic strong performance on Humanoid-Walk or Dog-Walk, any failure of our method to learn text-conditioned behavior in such environments could potentially be attributed to the underlying model rather than our proposed reward computation scheme. We selected TD-MPC because it has strong base performance on complex benchmarks (e.g. it is the first approach to even solve the default Dog task), and can therefore clearly showcase the difference between distinct text-to-reward computation methods **on top of it**.
> > >
> > > Nevertheless, we have discovered in early iterations of our method that TADPoLe is general across RL backbones. We have implemented RPO with TADPoLe (using the CleanRL implementation, where we replace the reward with the text-conditioned TADPoLe reward, and keep all default hyperparameters) and have achieved competitive performance on OpenAI Gym tasks:
> > > |Task|Prompt|TADPoLe|Ground-Truth|
> > > |:-|:-:|:-:|-:|
> > > |Hopper-v3|“a one-legged stick figure jumping forward over a grid floor”|2925.56|3852.78|
> > > |Walker2d-v4|“a stick figure walking across a grid floor”|2038.06|3872.54|
> > >
> > > Note that TADPoLe is not trained on ground-truth environmental signals, but still manages to achieve high ground-truth returns despite being optimized purely to align with a provided text prompt. We agree that more explorations on other RL backbones would further support the generalizability of TADPoLe, beyond the generalization we have already shown over different tasks (specified by flexible text prompts), as well as robotic states and environments (Humanoid, Dog, MetaWorld).
> > >
> > > **On real-world experiments:** We would like to clarify a potential misunderstanding on the "real-world experiments" presented in Diffusion-Reward: Diffusion-Reward *does not train any policies for real-world robots*; the only policies they train are in simulation, which we also do in this work. Instead, they use real-world videos to visualize their computed rewards in an *offline manner*. They compare reward curves between an expert and a random policy (as Diffusion-Reward does not have text-conditioning capabilities) to “indicat[e] the **potential** of [their] method for real-world robot manipulation tasks.”
> > >
> > > Analogously, for the reviewer's interest, we also visualize TADPoLe rewards for natural videos. As we are now interested in text-conditioned rewards, we compare between different text prompts for a given video demonstration, and show that the more-aligned text prompt has higher predicted reward. This is akin to what we have previously reported in Appendix Figure A1, but now for natural videos. Similar to what is reported in Diffusion-Reward, we visualize our computed dense reward for some real-world robotic arm demos from the Bridge Dataset (Diffusion-Reward did not release their robot-arm videos publicly, nor are they text-annotated). We also visualize some human actions. We provide the updated graph comparisons, along with the videos, on the [submission website](https://sites.google.com/view/tadpols/home), and verify that TADPoLe can determine if a natural video is more aligned with a provided text prompt or not, thus further supporting its use as a text-conditioned reward signal for policy learning. With this, we hope the reviewer increases their perception of our contribution to the community.

---

### Author Rebuttal · Authors · 2024-08-07

We thank all reviewers for their constructive feedback, and are glad that TADPoLe was recognized as “novel”, “easy to follow”, and “demonstrates versatility across different environments and tasks”.

We have identified common points raised by the reviewers, which we summarize and respond to below:

**Comparison with other diffusion-based rewards:** To the best of our knowledge, there is no prior work on text-aware diffusion rewards for an apple-to-apple comparison; as pointed out by reviewer TWnR, Diffusion-Reward is a relevant work (and also concurrent, as it is to be published at ECCV 24).  Furthermore, Diffusion-Reward does not take text conditioning, and requires the diffusion model to be trained with expert video demonstrations. TADPoLe, in comparison, is zero-shot and directly computes text-aware rewards through a generally pre-trained diffusion model without using any in-domain examples.  For the sake of comparison, however, we adapt Diffusion-Reward to be text-conditioned, and use the same diffusion model as TADPoLe for an apple-to-apple comparison on Meta-World.  We report the results in Table A8, found in the attached rebuttal Appendix page, and we discover that TADPoLe still outperforms Diffusion-Reward in overall task performance.

**Qualitative evaluation:**  Reviewers were interested in details of our qualitative evaluation, given the potential subjectivity introduced by this evaluation scheme.  We tried our best to make the human evaluation as impartial as possible; we performed our user study with 25 random participants through an anonymized random platform (Prolific) without prior training to estimate a general response from the human population.  We also use the videos of policy behavior at the last timestep of training without cherry-picking. In the attached rebuttal Appendix page, we provide the fine-grained user study results on what percentage of the users believe the video achieved by the policy is appropriately text-aligned.  We find that TADPoLe consistently achieves superior text-alignment preferences.
Robustness of hyperparameters: We highlight that although there are additional hyperparameters, which may require tuning, these hyperparameters can generally be shared across environments (Humanoid, Dog, MetaWorld), tasks (standing, kneeling, walking, etc.), and even diffusion models without modification (TADPoLe, Video-TADPoLe).  The selection of noise level, as well as hyperparameters $w_1$ and $w_2$ have been previously justified in the Appendix of the original submission.  We offer a new study on the symlog operation, and its effect on performance across tasks and environments, in Table A6 of the attached rebuttal Appendix page, and invite the reviewers’ interest towards it.

**Computational Cost:** we clarify that the computational cost of TADPoLe is simply one denoising forward step of a pretrained diffusion model to generate each dense reward. In comparison, Diffusion-Reward requires multiple denoising steps to generate one singular reward (in practice, 10 steps are used). TADPoLe is therefore computationally cheaper and faster, when the same diffusion model is utilized, in terms of reward computation.

We thank the reviewers for their time and consideration,

The Authors

---

### Decision · Program_Chairs · 2024-09-25

**Decision:**

Accept (poster)

**Comment:**

This paper has mixed scores of (4,4,6,7), with a variety of points made by the reviewers about its strengths and weaknesses. Overall, I have some doubts about some of the crucial points made by a few of the low-scoring reviews for a few reasons, finding that the main concerns were either resolved or not in conflict with the main contributions of the paper. I generally agreed with the remaining points made (positives and negatives alike). Taken together, I recommend acceptance, and for the authors to ensure that they address the remaining concerns of the reviewers. Below, I summarize some points made by reviewers that I used to arrive at this recommendation.

## Reviewer z3di (4)
This reviewer was unconvinced by the author responses, and stated:
> I don't think the videos shown by the authors are convincing ... The results using DrM on Adroit are also confusing.

and

> The authors said that 'prior work on MetaWorld mainly use dense ground-truth rewards, in-domain expert demonstrations, or both'. However, they mention DrM in their other replies, which uses sparse rewards and achieves much better performance."

I share these concerns, but find them minor, because they are not in conflict with the main claims of the paper ("our method is novel in its reward computation, as well as its utilization of a domain-agnostic generative model" ... "first approach to leverage domain-agnostic visual generative models for policy learning."). Furthermore, earlier in the review process, this reviewer stated

> As for the reason for real-world experiments, diffusion-reward actually added a lot of real-world experiments. Without real-world experiments, I think the contribution to the community is very limited. So I decide to maintain my score.

The authors responded to this criticism by pointing out "the referenced paper does not train any policies for real-world robots; the only policies they train are in simulation, which we also do in this work." I therefore consider this concern addressed, which was a major factor why this reviewer maintained their (4) score when they stated this concern.

## Reviewer iiYV (6)
This reviewer stated "Although I think this work still requires heavy prompt engineering and the generalization ability is weak, I would raise the score to 6."

## Reviewer TWnR (4)
This reviewer, when nudged for their final evaluation, stated there are still "questions about their work's performance and the true source of the observed performance improvement." This concern stems from the ablation study that demonstrated the importance of the 'symlog' transformation. However, this finding is consistent with the paper's claims ("our method is novel in its reward computation, as well as its utilization of a domain-agnostic generative model" ... "first approach to leverage domain-agnostic visual generative models for policy learning.") and in my opinion shouldn't count significantly against it. One issue I have with this finding, though, is that the current presentation of the paper does not make clear that it was discovered that the symlog operation was important as shown by Table A6 in the author's response, so it needs to be updated to reflect this finding. Furthermore, Table A6 is not a complete ablation across all environments and prompts. A more complete Table A6 would offer more evidence of this finding, and I strongly recommend the authors perform a more complete ablation. Whether additional ablations would support or undermine this finding it is irrelevant to the paper's main claims, and thus I discount this criticism a fair bit.

## Reviewer KsL9 (7)
This reviewer stated a mostly positive assessment with few minor weaknesses.